


# The Cambodian Mekong floodplain under the future development plans and climate change

Sokchhay Heng[1], Alexander Horton[2], Panha Hok[1], Sarit Chung[1], Jorma Koponen[3], Matti Kummu[2]

[1]Faculty of Hydrology and Water Resources Engineering, Institute of Technology of Cambodia, Phnom Penh, Cambodia
[2]Water and Development Research Group, Aalto University, Espoo, Finland
[3]EIA Finland Ltd., Espoo, Finland

*Correspondence to*: Sokchhay Heng (heng_sokchhay@yahoo.com)

**Abstract.** Water infrastructure development is crucial for driving economic growth in the developing countries of the Mekong. Yet it may also alter existing hydrological and flood conditions, with serious implications for water management,
agricultural production and ecosystem services, especially in the floodplain regions. Our current understanding of the hydrological and flood pattern changes associated with infrastructural development still contain several knowledge gaps, such as the consideration of overlooked prospective drivers, and the interactions between multiple drivers. This research attempts to conduct a cumulative impact assessment of flood changes in the Cambodian part of the Mekong floodplains. The developmental activity of six central sectors (hydropower, irrigation, navigation, flood protection, agricultural land use and
water use) as well as climate change were considered in our modelling analysis. Our results show that the monthly, sub-seasonal, and seasonal hydrological regimes will be subject to substantial alterations under the 2020 planned development scenario, and even larger alterations under the 2040 planned development scenario. The degree of hydrological alteration under the 2040 planned development is somewhat counteracted by the effect of climate change, as well as the removal of mainstream dams in the Lower Mekong Basin and hydropower mitigation investments. The likely impact of decreasing
water discharge in the early wet season (up to –34%) will pose a critical challenge to rice production, whereas the likely increase in water discharge in the mid-dry season (up to +54%) indicates improved water availability for coping with drought stresses and sustaining environmental flow. At the same time, these changes would have drastic impacts on total flood extent, which is projected to decline up to –18%, having potentially negative impacts on floodplain productivity whilst at the same time reducing the flood risk to the area. Our findings urge the timely establishment of adaptation and mitigation
strategies to manage such future environmental alterations in a sustainable manner.

## 1 Introduction

The Mekong River Basin (MRB) is the largest river basin in the Southeast Asian mainland. Historically, cyclones and severe tropical storms have generated the most significant Mekong flooding events, the largest of which was recorded in 1966, when tropical storm Phyllis struck the Upper Mekong Basin (UMB) (Adamson et al., 2009). At the downstream end of the
basin, severe floods have most commonly been recorded in the area around Stung Treng Province, Cambodia, at the





confluence of the Mekong River, and within the Mekong Delta. The last severe flood occurred in 2011 and it is ranked among the highest discharge recorded in the Lower Mekong Basin (LMB) (MRC, 2011).

Whilst prolonged flooding damages infrastructure, crops and floodplain vegetation, and the fertile land; annual flooding is a vital hydrological characteristic of the MRB, as it improves water availability during the dry season, and maintains and
increases the high productivity of ecosystems and biodiversity (Arias et al., 2014; Lamberts, 2008; Ziv et al., 2012; Arias et al., 2012; Boretti, 2020; Kondolf et al., 2018; Kummu et al., 2010; Kummu and Sarkkula, 2008; Schmitt et al., 2018; Schmitt et al., 2017; Västilä et al., 2010). As part of the annual flood cycle, floodwaters transport essential sediments and nutrients from the river channel into the floodplain, and distribute them across a wide area, fertilizing agricultural lands and enhancing floodplain productivity (Arias et al., 2014; Kummu and Sarkkula, 2008; Lamberts, 2008). Moreover, the wider the flood
extent, the larger the area of interaction between aquatic and terrestrial phases, which increases the potential transfer of floodplain terrestrial organic matter and energy into the aquatic phase. Under the combined impacts of hydropower and climate change, the flooded area in Cambodia's Tonle Sap Lake Basin is projected to decline by up to 11%, which may lead to a decline in the net sedimentation and the aquatic net primary production of up to 59%, and 38% respectively (Arias et al., 2014; Lamberts, 2008).

Existing hydrological and flood regimes will likely be altered due to climate change and infrastructure developments; but the degree of alterations vary with different drivers, location and time (Hoang et al., 2016; Hoang et al., 2019; Lauri et al., 2012; Piman et al., 2013; Try et al., 2020a). Hoang et al. (2016) projected that the Mekong's discharge under climate change conditions will decrease in the wet season (up to –7%), and increase in the dry season (up to +33%), equivalent to an annual increase between +5% and +15%. Lauri et al. (2012) pointed out that hydrological conditions of the MRB were
highly dependent upon the Global Climate Model (GCM) being used, with projections of water discharge at Kratie station, Cambodia, ranging from –11% to +15% for the wet season and from –10% to +13% for the dry season. The study also concluded that the impact on water discharge due to planned reservoirs was much larger than those simulated due to climate change, with water discharge during the dry and early wet season being primarily determined by reservoir operation. Hoang et al. (2019) found that hydropower development plans in MRB are expected to increase dry seasons flows up to +133% and
decrease wet season flows up to –16%. Acting in opposition to climate change, the future expansion of irrigated lands in the wider Mekong region is expected to reduce river flows up to –9% in the driest month (Hoang et al., 2019). These hydrological alternations are likely to intensify when considered cumulatively.

Changes to the Mekong mainstream flows will have direct impacts on flooding in the LMB floodplains in Cambodia and Vietnam. In the LMB part of Cambodia, Try et al. (2020a) projected an increased peak inundation area of 19–43% due
to climate change. Infrastructure development, in contrast, is expected to cause a decline in the Tonle Sap's flood extent by up to 1,200 km$^2$ (Arias et al., 2012), as dam development alone is expected to reduce flooded area in the Mekong Delta by 6% in the wet year and by 3% in the dry year (Dang et al., 2018). Flood extent in the Vietnam's Mekong Delta is projected to increase by 20% under the cumulative impacts of climate change and infrastructure development, bringing prolonging submergences of 1–2 months (Triet et al., 2020).





The impacts described above may eventually lead to a new hydrological and flood regime in the Mekong region, and would likely endanger the riverine ecology and endemic aquatic species of the Mekong floodplain (Arias et al., 2012; Dang et al., 2018; Kummu and Sarkkula, 2008; Räsänen et al., 2012). To effectively manage and overcome these pressures and challenges in any particular floodplain, there is an urgent need to evaluate the combined impacts of climate change and infrastructure operations basin-wide (Västilä et al., 2010; Hoang et al., 2019; Hoanh et al., 2010; Lauri et al., 2012).

However, the existing studies have focused either on the basin scale flow changes (Dang et al., 2018; Hoang et al., 2016; Hoang et al., 2019; Hoanh et al., 2010; Lauri et al., 2012; Pokhrel et al., 2018; Try et al., 2020a) or assessed the impacts on flooding either for the Tonle Sap (Arias et al., 2012; Ji et al., 2018; Yu et al., 2019) or Vietnamese parts of the Mekong Delta (Dang et al., 2018; Tran et al., 2018; Triet et al., 2020). Very little is known how basin-wide development and climate change would impact Cambodian floodplains other than Tonle Sap (Fig. 1), despite them being important agricultural lands

and home to more than 6.4 million people (2008 Population Census).

     Therefore, we have attempted to quantify the cumulative impacts of water resources development plans and climate change on hydrological and flood conditions localised in the Cambodian Mekong floodplain (Fig. 1) by using state-of-the-art hydrological and hydrodynamic models. In concentrating on the provincial level, using an extended time-series for the calibration period, validating the flood extent against satellite imagery, and incorporating a larger set of driving factors

within our analysis, the present study is a novel and important contribution to the work being done to understand the potential for future changes to the complex hydrology of the Mekong floodplain in Cambodia. The results of this study are crucial for proposing and formulating adaptation and mitigation strategies to the flood-prone areas, identifying the main drivers causing floods at the provincial level for better flood management, and supporting the government in meeting the national and global sustainable development goals.

## 2 Materials and methods

### 2.1 Study area

The study area is located in the downstream part of the Mekong River Basin (excluding the Tonle Sap Lake region), also known as the "Cambodian Mekong floodplain" (Fig. 1). The area is about 27,760 km$^2$ and extends along the Mekong mainstream from Kratie province to the Cambodia-Vietnam border. It covers parts of 12 provinces in Cambodia and one

province in Vietnam (Tay Ninh).

     A major part of the Cambodian Mekong floodplain is characterized by a flat terrace and low-lying grounds with gentle slopes that contain many depressions and lakes, except for the upper parts of the Prek Thnot and Prek Chhlong tributaries, which contains steeper terrain. Conditions within the area are dominated by the seasonality and year-to-year variability of the Mekong flow regimes. During the flood season, the characteristics of the floodplain and Tonle Sap Lake play a vital role in

flood peak attenuation and regulation temporarily storing and later conveying water across the vast low-lying areas. During the wet season, water flows from the Mekong mainstream into the Tonle Sap Lake, but this flow is then reversed in the dry





season. This illustrates the highly complex hydrological system at play throughout the region, and the extreme seasonal variations that characterize the ecological and agricultural landscape.

Within our historic baseline period of 1985–2008, the catchment annual average temperature varies from 27.2°C to
100 28.3°C, with mean monthly temperatures between 30°C during the hottest months (April and/or May), and 26°C in the coldest month (January). Average annual rainfall in the Cambodian Mekong floodplain during the same period varies between 1,100 mm and 1,850 mm, with mean monthly rainfall ranging between 250 mm in the wettest months, and 10 mm in the driest.

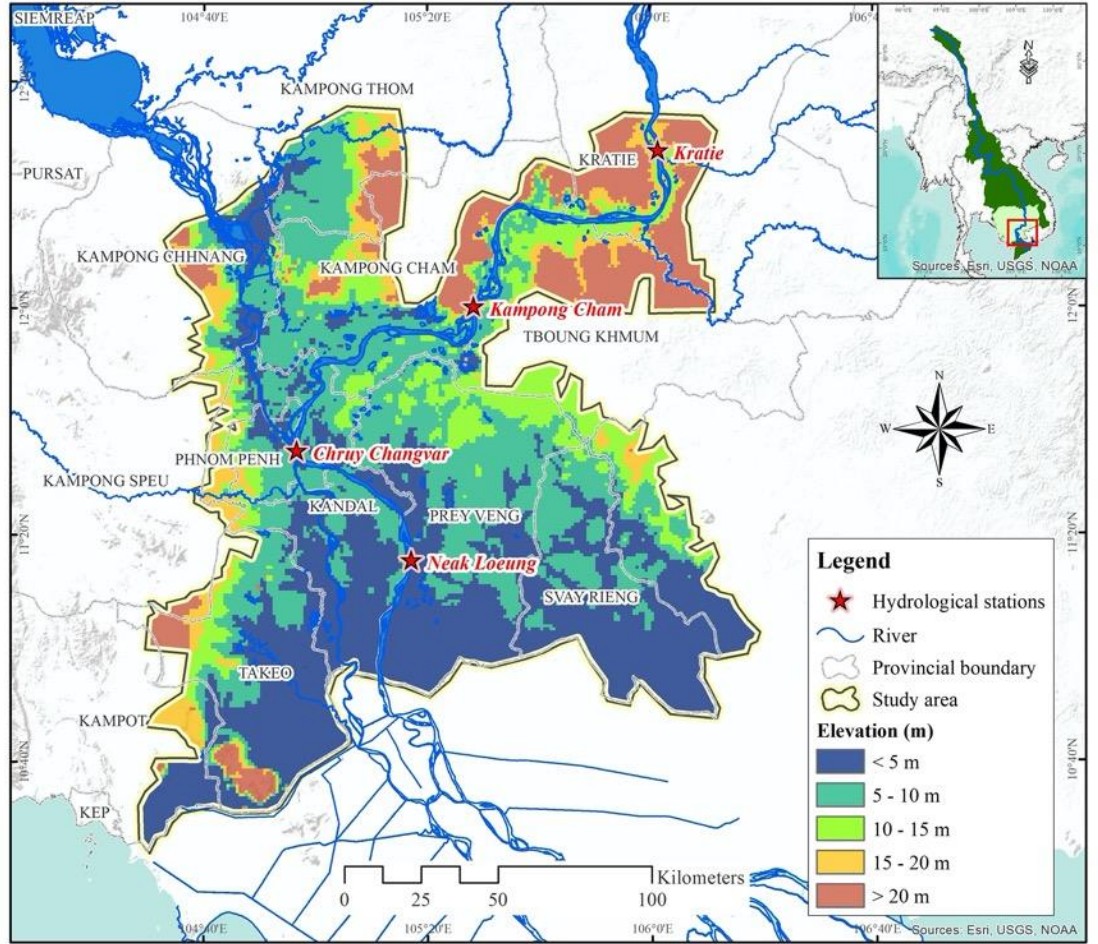

**Figure 1. Map of the study area, the Cambodian Mekong floodplain. Elevation of 90-m grid cell was extracted from the SRTM database and river lines were obtained from the MRC database.**





## 2.2 Datasets

We used an existing distributed hydrological – floodplain model combination, consisting of IWRM-VMod and the
110 floodplain model IWRM-Sub (MRC, 2018a). Constructed for the MRC's Council Study by Jorma Koponen and his team,
the models are based on the SRTM 90-m topographical map (Jarvis et al., 2008), soil types map (FAO, 2003), and land use
map (GLC2000, 2003), all aggregated to 1 km × 1 km resolution. The daily meteorological and hydrological data for the
period 1985–2008, geospatial data, and river cross-section data were retrieved from the Mekong River Commission (MRC).
The satellite images of Landsat 5 were used to generate the flood extent maps based on a sophisticated water detection
algorithm developed and optimized for the Lower Mekong region (Donchyts et al., 2016). The additional boundary
conditions of Mekong River inflow at Kratie and Tonle Sap Great Lake were obtained from the MRC Decision Support
Framework model comprising of the Soil and Water Assessment Tool (SWAT) and Integrated Quantity and Quality Model
(IQQM). For the initial condition of the floodplain hydrodynamic model, flood points (water level) generated from the
hydrodynamic model (ISIS) were also obtained from MRC. All model inputs and their brief description are presented in
Table 1.

**Table 1. List and brief description of datasets.**

| No. | Data type | Period | Resolution | Source |
|---|---|---|---|---|
| 1 | Topography (digital elevation model) | – | 90 m | Shuttle Radar Topography Mission |
| 2 | Land use map | 2003 | 1 km | Global Land Cover 2000 |
| 3 | Soil types map | 2003 | 1 km | Food and Agriculture Organization |
| 4 | Meteorological data | 1985–2008 | Daily | Mekong River Commission |
|  | • Temperature | | | |
|  | • Rainfall | | | |
| 5 | Hydrological data | 1985–2008 | Daily | Mekong River Commission |
|  | • Discharge | | | |
|  | • Inflow | | | |
|  | • Floodpoints | | | |
| 6 | Geospatial data | – | – | Mekong River Commission |
| 7 | Hydropower dams and irrigation | – | – | Mekong River Commission |
| 8 | Climate change (mean warmer & seasonal) | 1985–2008 | Daily | Mekong River Commission |
| 9 | Flood extent maps (satellite image) | 1985–2008 | 30 m | SERVIR-Mekong |
| 10 | River cross-section | – | – | Mekong River Commission |

## 2.3 Modelling methodology

We applied the existing IWRM-VMod and IWRM-Sub models to assess the cumulative impacts of future development plans
and climate change on the Cambodian Mekong floodplain (Hoang et al., 2019; MRC, 2018a; ICEM and Alluvium, 2018;
Räsänen et al., 2012). Here we attempt to enhance the reliability of these existing models, particularly in the Cambodian



Mekong floodplain, by advancing the predictive accuracy of the hydrology (recalibration), accounting for multiple calibration stations (four stations), and validating flood extents against satellite imagery; as described below.

Our initial model setup describes the current state of the floodplain for the historic baseline period of 1985–2008, which we calibrated and validated against observations of water discharge and water level taken at Kratie, Kampong Cham, Chruy

Changvar, and Neak Loeung hydrological stations (see locations in Fig. 1). The model performance was systematically quantified and evaluated based upon: Nash-Sutcliffe efficiency (NSE), percent bias (PBIAS), ratio of the root mean square error to the standard deviation of observed data (RSR), and coefficient of determination ($R^2$). For the range adopted for performance rating see ASABE (2017).

Flood extent maps generated from the IWRM-Sub model were validated for the same period against satellite-based

flood extent maps generated by the Surface Water Mapping Tool (SWMT; see below for more information). To evaluate the model performance for flood inundation maps, we applied three indices: hit ratio (HR), true ratio (TR), and the normalized error (NE). HR evaluates how much of the flood derived from remote sensing images are identified by the simulation. TR evaluates how much of the simulated extent agrees with the remote sensing. NE evaluates the relative errors in the total area of flood extents. If both estimations overlap the area perfectly, both TR and HR become 1 and NE becomes 0.

Once the model was successfully calibrated and validated, we modulated the inflow at Kratie and Chruy Changvar stations to represent the upstream impacts of various development and climate change scenarios (see Sect. 2.4). We then simulated the Cambodian Mekong floodplain's hydrological and flood conditions (flood extent, flood depth, and flood duration) for each scenario. The overall methodological framework adopted in this study is depicted in Fig. S1.

For each time step, the Cambodian Mekong floodplain model (combination of IWRM-VMod and IWRM-Sub models)

first interpolates meteorological data for each grid cell from observation point data using a height correction factor where required (ICEM and Alluvium, 2018). In addition, initial and boundary conditions (flood points and inflow) were incorporated into the model structure. To produce an initial flood extent map, we extracted flood points (water level) from the ISIS model.

The Surface Water Mapping Tool (SWMT) is a Google Appspot based online application developed by Donchyts et al.

(2016) with the full support by the SERVIR-Mekong project for the Mekong River Basin. A stack of Landsat 8 (also including 4, 5, and 7) data was generated using SWMT from the present period back to 1984. From this stack of images, two percentile maps were calculated, which represent two different situations: a permanent situation of the higher percentile (default value of 40) and a temporary situation of the lower percentile (default value of 8). Xu (2006) used the Modified Normalized Difference Water Index (MNDWI) to quantify the water index map from these percentile maps. Several spectral

bands from the Landsat satellites were combined using the MNDWI, which are sensitive to the occurrence of water. Then the water and non-water areas can be classified from the threshold value applied to each pixel level. To improve the results, some corrections were performed to minimize errors associated with falsely classified water over dark vegetation and (hill) shadows. Dark vegetation is masked out using the Normalized Difference Vegetation Index (NDVI) and hill shadows are masked out using a Height Above Nearest Drainage (HAND) map (Rennó et al., 2008), derived from the Multi-Error-





Removed Improved-Terrain (MERIT) Digital Elevation Model (DEM) (Yamazaki et al., 2017). Figure S2 illustrates all
procedures of the Surface Water Mapping Tool.

## 2.4 Analytical scenario descriptions

The scenario setups of this study are almost identical to scenarios from the MRC's Council Study consisting of three main
water resource development scenarios. The baseline conditions represent year 2007 situation (BASE scenario). The medium-
165 term development scenario is for the definite future of 2020 (Def2020). The long-term development scenario is for the
planned development in 2040 (Pla2040). On top of these, there are three other sub-scenario setups, which are variations of
the 2040 planned development (Table 2). Figure S3 shows the overall list of employed hydropower dams within the Mekong
basin (MRC, 2019). The hydropower development scenario consists of 126 dams on both mainstreams (16) and tributaries
(110), according to the compiled database from ADB (2004) and the Mekong River Commission (MRC, 2009). Further
information related to these hydropower dams' characteristics and names can be found in MRC (2018b, 2016).

The BASE scenario includes 2007 LMB tributary and China mainstream hydropower dams (Manwan and Dachaoshan
only), agricultural land use, irrigation schemes, water navigation, flood protection, as well as domestic and industrial water
use. It represents the baseline conditions in the LMB used to compare against all other scenarios. The Pla2020 scenario
(medium-term development) includes 2020 LMB tributary, LMB mainstream (Xayaburi and Don Sahong only) and China
mainstream hydropower projects (11 dams), agricultural land use, irrigation schemes, water navigation, flood protection, as
well as domestic and industrial water use. The Pla2040 scenario (long-term development) consists of LMB tributary, LMB
mainstream (11 dams) and China mainstream hydropower projects (12 dams), as well as the aforementioned agricultural
land use, etc. The Pla2040CC is the same development setup as Pla2040, but with climate change incorporated into the
projection (IPSL-CM5A-MR under RCP4.5). Based on the IPCC's approach, the GCM selected for this study under the
180 medium emission scenario represents the range of uncertainty inherent in the GCM climate change projections for the LMB
(MRC, 2017), as it is characterized by an increased seasonal variability (wetter wet and drier dry seasons),and covers the
monsoon seasonality (Her et al., 2019). There are then two additional sub-scenarios adapted from Pla2040; the
Pla2040NoHPP scenario (2040 plans, LMB tributary and Chinese mainstream dams, but without LMB mainstream dams)
and the Pla2040MiHPP scenario (2040 plans, mitigation measures and joint operation of key dams) (MRC, 2019). The
185 mitigation measures and joint operation of key dams denote a good coordination among all mainstream hydropower dams;
their operation is equipped with measures for navigation lock, fish passage, sediment flushing, environmental flow, and
water quality maintenance (MRC, 2020). For more information related to hydropower development and irrigation scenarios
see Hoang et al. (2019).





**Table 2. Summary of scenario considerations.**

| Scenario | | Level of development for water-related sectors* | | | | | | Climate | Floodplain settlement |
|---|---|---|---|---|---|---|---|---|---|
| | | ALU | DIW | FPF | HPP | IRR | NAV | | |
| BASE | Baseline | 2007 | 2007 | 2007 | 2007 | 2007 | 2007 | 1985–2008 | 2007 |
| Def2020 | Definite future scenario 2020 | 2020 | 2020 | 2020 | 2020 | 2020 | 2020 | 1985–2008 | 2020 |
| Pla2040 | Planned development scenario 2040, no climate change | 2040 | 2040 | 2040 | 2040 | 2040 | 2040 | 1985–2008 | 2040 |
| Pla2040CC | Planned development scenario 2040 with climate change | 2040 | 2040 | 2040 | 2040 | 2040 | 2040 | Mean warmer & wetter | 2040 |
| Pla2040NoHPP | Planned development 2040 without mainstream HPP | 2040 | 2040 | 2040 | Only tributary 2040 | 2040 | 2040 | 1985–2008 | 2040 |
| Pla2040MiHPP | Planned development 2040 with HPP mitigation investments | 2040 | 2040 | 2040 | 2040 | 2040 | 2040 | 1985–2008 | 2040 |

*ALU = Agriculture/Land use change; DIW = Domestic and Industrial Water Use; FPF = Flood Protection Infrastructure; HPP = Hydropower; IRR = Irrigation; and NAV = Navigation

# 3 Results

## 3.1 Predictive accuracy of the model

Based on the validation measures (Table 3), a good model performance is obtained at all stations (both water discharge and
195 water level) with the values of NSE between 0.62 and 0.96, PBIAS between –3.68% and +20.66%, RSR between 0.19 and 0.45, and $R^2$ between 0.89 and 0.97. It should be noted that the statistical model performance with NSE and $R^2$ greater than 0.5, PBIAS between ±25%, and RSR less than 0.7 is indicated as decision guidelines for hydrologic model studies (Benaman et al., 2005; Setegn et al., 2010). A time series comparison between the simulated and observed water discharge and water level (1985–2008) at four hydrological stations can be found in Fig. S4. It is apparent that the simulated water discharge
among these stations is well in line with the observed data throughout the 24-year study period; however, three stations, namely Kampong Cham, Chruy Changvar, and Neak Loeung overestimate the peak water discharge and water level. The model consistently overestimates the medium and high water discharges at Neak Loeung station, and the overall predictive accuracy at this station is also lower than at the other three stations (Table 3). This could be due to the complex flow system between the Mekong and Tonle Sap River which cannot be fully captured by the model, especially for stations in the Lower
Mekong River downstream of the Phnom Penh junction.

Results of the flood extent map from the Cambodian floodplain model and SWMT (Landsat 5) observations over the time horizon 1985–2008 show a very good agreement. However, the model does overestimate the flooded area by about 14%, with the overlapping flooded area being about 11,640 km$^2$ (73% of the IWRM-Sub model area and 84% of the SWMT


area) (Fig. 2). The overestimation could be attributed to the use of a large spatial resolution by the model (1 km × 1 km)

while the satellite data is at a 30-m spatial resolution. Moreover, a lot of scattering in the flood extent is noticeable from the generated satellite image. Nevertheless, both results look very promising as indicated by the three evaluation indices (HR = 0.84, TR = 0.73, and NE = 0.14).

**Table 3. Statistical model performance at four hydrological stations (1985–2008). See station locations in Fig. 1.**

| Station | Water discharge | | | | Water level | | | |
|---|---|---|---|---|---|---|---|---|
| | NSE | PBIAS (%) | RSR | $R^2$ | NSE | PBIAS (%) | RSR | $R^2$ |
| Kratie | 0.96 | 8.37 | 0.19 | 0.97 | 0.96 | –3.68 | 0.20 | 0.97 |
| Kampong Cham | 0.96 | 5.70 | 0.19 | 0.97 | 0.92 | 0.02 | 0.28 | 0.92 |
| Chruy Changvar | 0.88 | 20.66 | 0.34 | 0.93 | 0.92 | 0.32 | 0.31 | 0.92 |
| Neak Loeung | 0.62 | 19.38 | 0.45 | 0.89 | 0.85 | 10.23 | 0.34 | 0.91 |

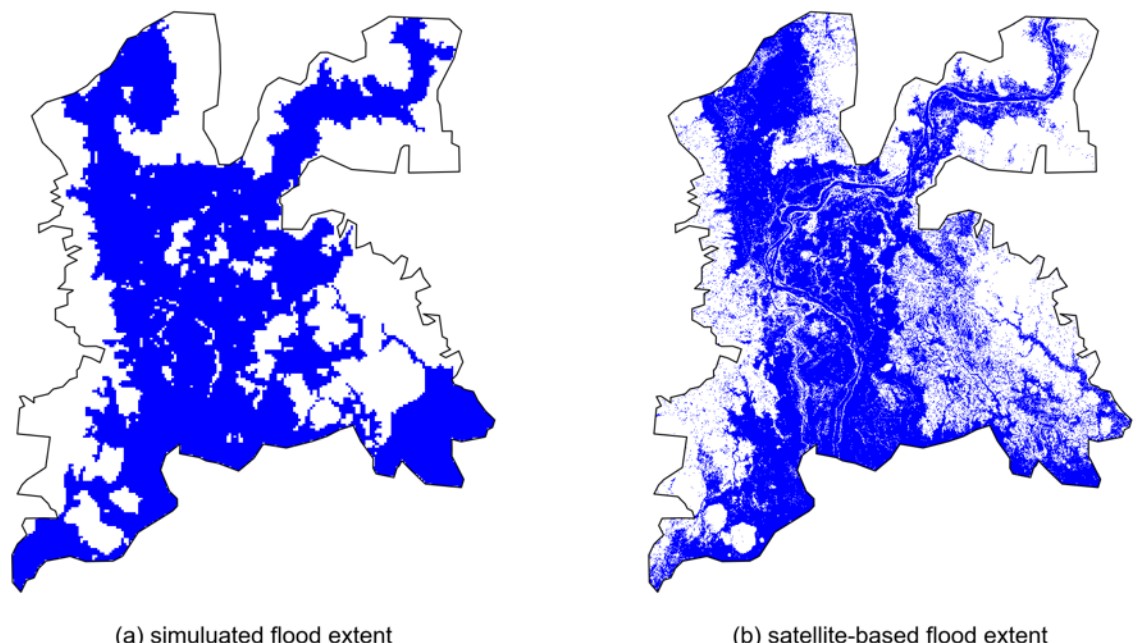

(a) simuluated flood extent          (b) satellite-based flood extent

**Figure 2. Comparison of maximum flood extent resulted from the model and satellite images (HR = 0.84, TR = 0.73, and NE = 0.14).**

### 3.2 Impacts on hydrological conditions

Having run the model for each of the development scenarios (BASE, Pla2020, Pla2040, Pla2040CC, Pla2040NoHPP, Pla2040MiHPP), we obtained the corresponding daily time series of water discharge and water level at each station, and

220 compared them with the baseline scenario (Fig. S5). We then calculated the flow duration curves for each scenario at each


station (Fig. S6), and the mean monthly water discharge and water level across the study period (Fig. S7). Finally, we computed the percentage change in mean monthly water discharge and water level for each scenario at each station (Fig. 3).

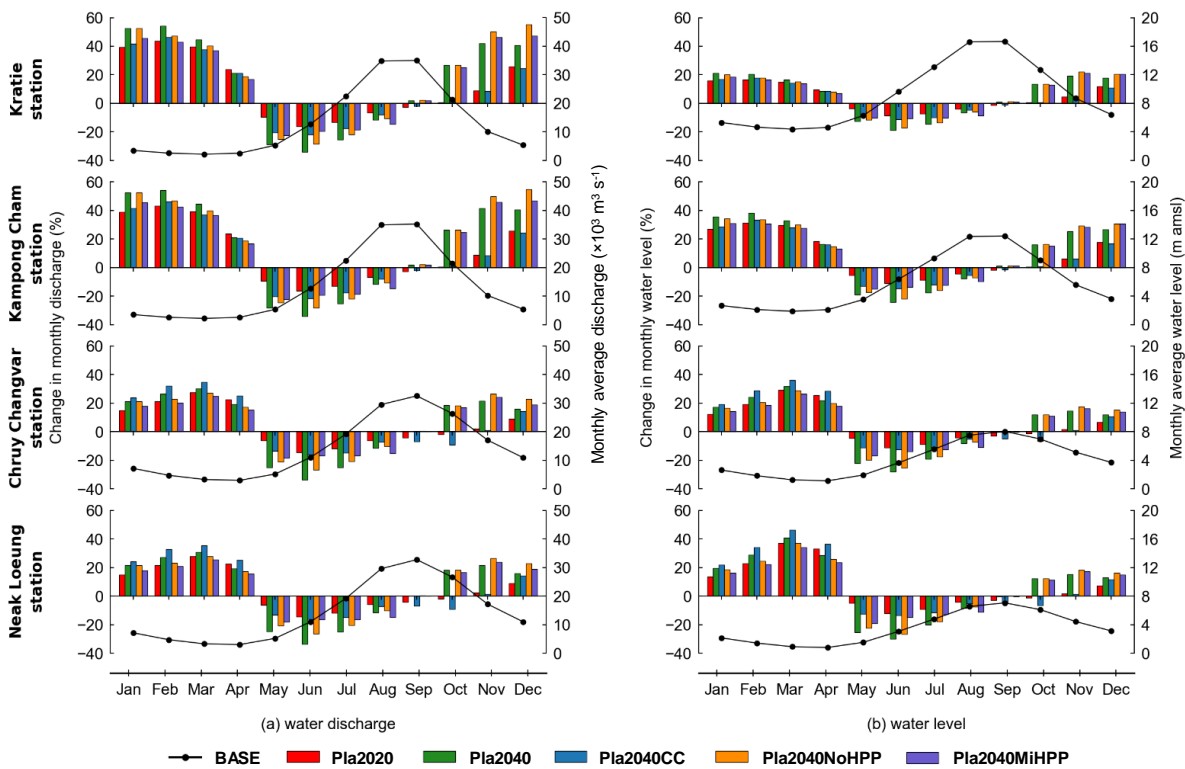

**Figure 3. Changes in monthly water discharge and water level at four monitoring stations; the black line with markers indicates the monthly water discharge and water level; the colour bar chart indicates the percentage change under different scenarios in comparison with the baseline (1985–2008).**

All scenarios follow the same generic pattern of increasing both water discharge and water level during the dry season (Nov–Apr), whilst reducing water discharge and water level during the early and mid- wet season (May–Aug). The late wet season (Sep–Oct) is characterized by a mixed pattern of changes (increasing and decreasing). The degree of alteration to these hydrological indicators is most pronounced in the upstream area of Kratie station and diminishes downstream towards Neak Loeung station. February and March display the highest magnitude of alterations to the wet season water discharge and water level increases, while June displays the largest decrease in dry season flows and water levels.

Comparing scenarios Pla2020 and Pla2040 illustrates the incremental impact of future planned developments throughout the Mekong independently of climate variability, as the same climate data was used for both scenarios. Across all four stations, and throughout the years, the proportional impact of Pla2040 is significantly larger than for Pla2020, especially in the wet and early dry season months (May–Dec). This demonstrates that the planned development for the 2020–2040 period will severely impact the hydrological functioning of the Mekong main channel, raising the dry season flows and reducing the wet season flows to slightly homogenise the river's hydrograph. The expected mean dry season flows increase by up to





+50% (in January and February) at upstream stations, and reduce wet season flows by more than –30% (in June) at all
stations.  The incorporation of climate change into the Pla2040CC scenario reverses the magnitude of these developmental
impacts between May and December, as the warmer dry season months and wetter wet season months compensate for the
anthropogenic flow alterations. Between January and April, the climate change impact is less consistent, showing opposing
trends at Kratie and Kampong Cham (upstream stations) compared to Chruy Changvar and Neak Loeung (downstream
stations). Though this may in part be due to model overestimation at downstream stations (Fig. S4).

Of all the scenarios, Pla2040NoHPP shows the largest proportional changes at the onset of the dry season (Nov–Dec),
slightly intensifying the proportional impact of developments compared to Pla2040. However, for the rest of the months
(during Jan–Oct) both Pla2040NoHPP and to a greater extent Pla2040MiHPP show a reduction in the proportional change in
both water discharge and water level compared to Pla2040. The difference between the changes shown in Pla2040 compared
to Pla2040NoHPP can be interpreted as the impact of developing mainstream dams in isolation; and comparing
Pla2040MiHPP with Pla2040 shows the impact of mitigation measures in isolation. Our results suggest that development
that excludes mainstream dams with the incorporation of mitigation investments would be the most sustainable in terms of
minimising hydrological alterations, which will be aided in this respect by the influence of climate change.

### 3.3 Impacts on flood conditions

Here we present the quantitative results together with the spatial analysis of flood conditions throughout the entire study
area. The comparisons between each scenario and their justifications are described in the analysis at the provincial level
because of the similarity in patterns. Under the baseline scenario (BASE), the modelling results between 1985 and 2008
show that the total flooded area ranges from 5,611 to 12,634 km$^2$. Its mean annual value is estimated at 9,477 km$^2$, about
34% of the whole study area.

The impact of planned development up until 2020 (Pla2020 scenario) is to reduce the total flooded area from the
baseline period in all years, with an average reduction of –6.3% (Fig. 4). This reduction is exacerbated by the planned
development of 2020–2040 (Pla2040) further reducing the total flooded area to an average of –7.9% compared to the
baseline period. However, the inclusion of climate change in the Pla2040 scenario (Pla2040CC) counteracts the
anthropogenic impact so that years that see reductions are lessened (mean = –5.3%), and some years see substantial increases
in the total flooded area (mean = +13.8%), with the average reduction across the entire study period being just –0.5% (Fig.
4). We see a similar pattern for development scenarios that exclude the mainstream dams without accounting for climate
change (Pla2040NoHPP), and include mitigation measures without accounting for climate change (Pla2040MiHPP), where
the years that see reductions in the total flooded area are less reduced (means of –4.3% and –4.6% respectively), and some
years see substantial increases (means of +14.1% and +9.8% respectively) with average changes in total flooded area of
+0.3% and +1.4% compared to the baseline period. If the impact of the mitigation measures incorporated into the
development scenario of Pla2040MiHPP were to combine with the impact of climate change evident in scenario Pla2040CC,
then the total flooded area might be expected to increase more substantially.



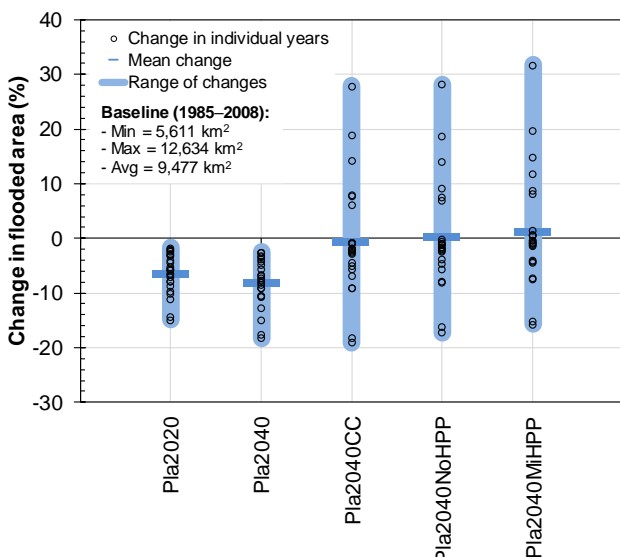

**Figure 4. Changes in total flooded area over the baseline period 1985–2008; the graph shows the range of changes due to interannual variation (rounded vertical bar), the value for mean change (horizontal line) and the values for change in individual**
**years (circles).**

The spatial distribution of flood inundation and depth across the Cambodian Mekong floodplain varies greatly between scenarios of planned developments and climate change (Fig. 5 and Fig. S8). The floodplain is characterized spatially by a high fluctuation of flood depth and flood duration alteration of over ±50% in all scenarios (Pla2020, Pla2040, Pla2040CC, Pla2040NoHPP, and Pla2040MiHPP), especially in the Southeast and the Southwest. Whilst the magnitude of these

280 fluctuations is large across all scenarios, it is most evident in scenarios Pla2020 and Pla2040, and less so in scenarios Pla2040CC, Pla2040NoHPP, and Pla2040MiHPP. Outside the hotspot areas (Southeast and the Southwest), the flood depth alteration varies between –20% and 0% under scenarios Pla2020 and Pla2040, and between –10% and +50% under scenarios Pla2040CC, Pla2040NoHPP, and Pla2040MiHPP. Our results suggest that hydropower dams would lower the flood depth, but the effect of climate change under wetter conditions would cause an increase in most areas which are currently prone to

285 flooding. In addition, the planned developments under Pla2020 and Pla2040 would likely reduce the flood duration between 10% and 20% in most areas, which might be seen as a benefit for flood protection measures in the region. By contrast, the scenarios Pla2040CC, Pla2040NoHPP and Pla2040MiHPP show a slight shift in the flood duration either increasing or decreasing by 10% across the majority of the study area, mainly in the low-lying areas along the Mekong River and its main tributaries. In summary, the planned developments of 2020–2040 will reduce both the flood depth and duration across most

290 of the floodplain, whilst the exclusion of mainstream dams, mitigation measures and climate change will have the opposite effect of increasing flood depths and durations, though these impacts are spatially heterogenic and highly variable.




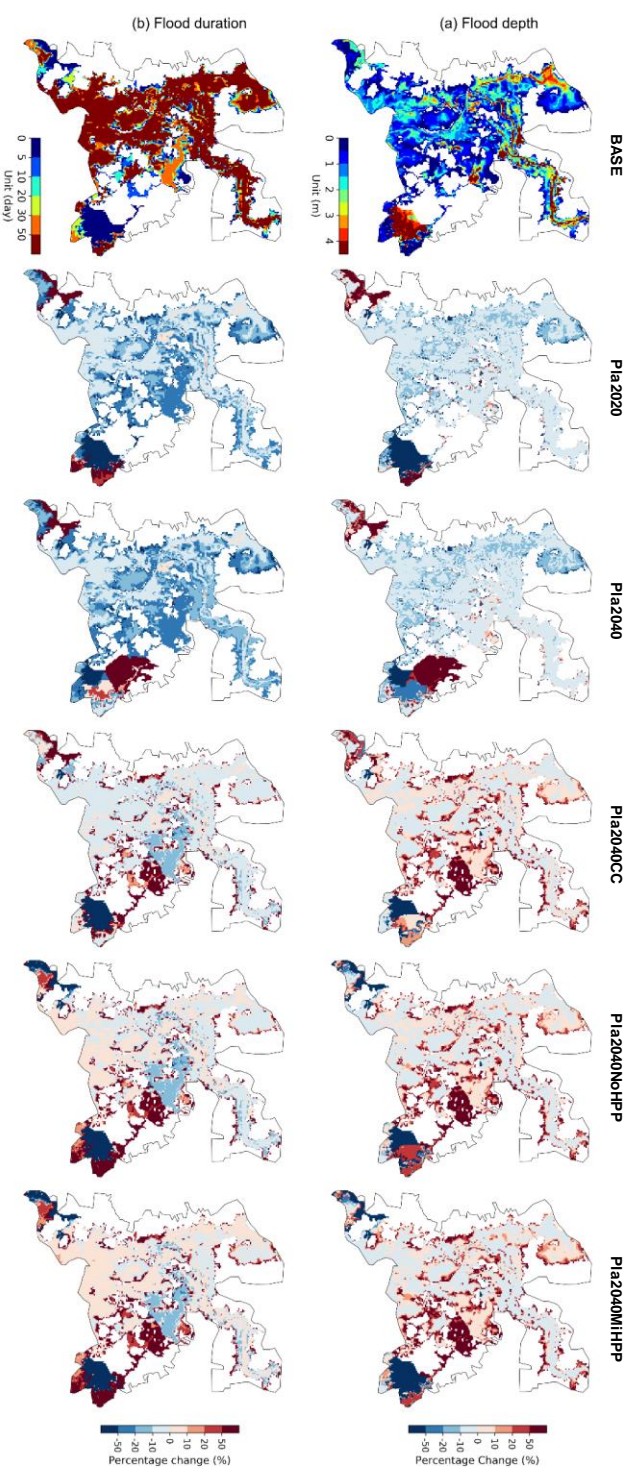

**Figure 5. Spatial distribution of changes in flood depth (upper row) and flood duration (lower row) over the baseline period 1985–2008, and all scenarios.**



## 3.4 Provincial level analysis

As the Cambodian Mekong floodplain covers only a little part of Kampong Speu and Kampot province, and Tay Ninh province is in Vietnam, we did not present results for these regions. For the remaining 10 provinces, we examined the change in flooded area, flood depth and flood duration for each scenario compared to the baseline period at the provincial level (Fig. 6). Here we present only the key results, with the detailed analysis being given in the Supplementary. Under the baseline scenario (BASE), the modelling results show that the average flooded area ranges from a minimum of 184 km$^2$ in Svay Rieng province to a maximum of 2,251 km$^2$ in Prey Veng province, which represents 43% of the provincial territory. Whilst the average flood depth ranges from 1.1 m in Svay Rieng province to 4.9 m in Kratie province, and the average flood duration ranges from 6 days in Svay Rieng province to 85 days in Kandal province. As a whole, Kampong Thom province receives the largest flood protection benefit from the planned developments between 2020 and 2040, with reductions under the Pla2040 scenario of –10.5% for flooded area, –6.0% for flood depth, and –14.1% for flood duration. Kampong Chhnang province receives the least benefit from such developments in terms of flooded area (only –3.1% under Pla2040) and flood duration (only –4.3% under Pla2040), while Kampong Cham province receives the least benefit in terms of flood depths, only –2.5% under Pla2040.





**Figure 6.** Changes in flooded area, flood depth, and flood duration over the baseline period 1985–2008 at provincial level. For Svay Rieng province, the scale of vertical axis is different from other provinces. For scenarios Pla2040CC, Pla2040NoHPP and Pla2040MiHPP, means of all 10 provinces are strongly controlled the large increases in Svay Rieng province. See province location in Fig. 1.



## 4 Discussion

### 315    4.1 Key findings

Our hydrological simulation accuracy of water discharge and water level for the baseline period of 1985–2008 at all four monitoring stations (Kratie, Kampong Cham, Chruy Changvar and Neak Loeung) exceeds existing studies within the same region (Hoang et al., 2016; Hoang et al., 2019; Västilä et al., 2010), with the possible exception of Dang et al. (2018), who recorded an NSE value of 0.98 compared to our value of 0.92 at Kampong Cham station. Nevertheless, the relative success

of our baseline simulations allows us to have great confidence in our future projections of hydrological responses within the bounds of error inherent within the GCM predations of future climate change.

Whilst there are individual studies of flood extent within our study region that slightly surpass our own in terms of accuracy when focusing on a single event (Fujii et al., 2003), our continual analysis of annual flood patterns comprising a 24-year time horizon is comparable to, and often exceeds, the accuracy of other such multi-year analyses done in the region

(Try et al., 2020a; Try et al., 2020b). This conformation of our initial baseline simulations of flood extent again suggests that we might have a high degree of confidence in our future projections of the Cambodian Mekong floodplain's response to changes in the flood hydrograph.

The future projections of all of the scenarios that we considered within our analysis, followed the same generic pattern of alterations to both the expected water discharge and river water level, increasing during the dry season (Nov–Apr), and

decreasing during the early- and mid- wet season (May–Aug). Such general pattern of alteration is due to the combined impacts of multiple drivers and the compensation between them. However, the late wet season (Sep–Oct) is characterized by a mixed pattern of changes (increasing and decreasing), which may be due to the uncertainty inherent in climate change simulations, and the effect of extreme flood events where inflow exceeds the flood storage capacity of reservoirs. These general trends are in line with the majority of previous researches in the region (Dang et al., 2018; Hoang et al., 2016; Hoang

et al., 2019; Lauri et al., 2012; Piman et al., 2013; Räsänen et al., 2012; Västilä et al., 2010). The degree of alteration to these hydrological indicators is most pronounced in the upstream area of Kratie station and diminishes downstream towards Neak Loeung station, which is also consistent with earlier findings (Dang et al., 2018; Lauri et al., 2012).

Our findings clearly demonstrate that the degree of hydrological alteration expected under the full development (Pla2040) scenario is diminished by the effect of climate change, and further reduced by the absence of mainstream dams in

the Lower Mekong Basin and hydropower mitigation investments. During the wet and early dry season (May–Dec), climate change would play the most important role in reducing the developmental impacts on hydrology, while during the mid- and late dry season (Jan–Apr), hydropower mitigation investments would be the most important driver counteracting developmental impacts. These findings support previous evidence that climate change may act in opposition to the impact of planned developments along the Mekong (Hoang et al., 2019).

The exclusion of LMB mainstream dams, by contrast, may contribute only slightly to counteracting developmental impacts, as the proposed LMB mainstream dams are mostly run-of-the-river types, with low height and little storage





capacity, which maintains the natural flow of the rivers to the benefit of ecosystem productivity but to the detriment of flood prevention efforts. Moreover, having its outlet upstream of Kratie station, the 3S basin contributes a large fraction of the Mekong's annual flows (20%) and consists of 42 dams in total (on-going and future development). The full development of
these 42 dams will lead to substantially increasing dry season flows (63%) and decreasing wet season flows (22%) (Piman et al., 2013). Such considerable impacts may already dominate downstream flow alternations, further reducing the potential impact of planned LMB mainstream dams.

Our future projections of flood conditions suggest that most provinces will see a decline in depth, duration, and area, reflecting the flood prevention benefit afforded by the Pla2040 scenario. However, as with the water discharge and water
level results, the impact of planned developments on flood prevention measures is counteracted to a large degree by the effect of climate change, absent mainstream dams in the Lower Mekong Basin, and hydropower mitigation investments. These findings are supported by our earlier results and previous studies that have concentrated on similar areas and drivers (Fujii et al., 2003; Pokhrel et al., 2018; Try et al., 2020a; Try et al., 2020b).

Our provincial level assessment shows that Prey Veng province is most vulnerable to the largest flooded area, as its
large territory is entirely located in the low-lying area adjacent to the Mekong River. Kampong Thom province receives the largest flood prevention benefit provided by the planned developments of 2020 and 2040. Kampong Chhnang province receives the least benefit from such developments in terms of flooded area and flood duration, most likely because the flood regime is controlled by the Tonle Sap Lake System and receives only a minor influence from the upstream flow alterations. Meanwhile, in terms of flood depth, Kampong Cham province receives the least benefit from such developments, as it
mainly functions as a transfer zone of the Mekong flood-flow from upstream to the floodplain and delta. Svay Rieng province is designated as the most vulnerable to the effect of climate change, as well as the province most effected by the reduction in flood protection benefit provided by the exclusion of LMB mainstream dams, and the adverse impact of mitigation investments. This is most likely due to the extremely low ground surface elevation (majority less than 8 m).

## 4.2 Implications of hydrological and flood condition changes

Changes in hydrological and flood conditions in the Cambodian Mekong floodplain could imply both positive and negative consequences to various sectors such as water resource management, agricultural productions, and ecosystem services (Arias et al., 2012; Kummu and Sarkkula, 2008). In addition, the direction, magnitude and frequency of impacts will be varied from one location to another.

The beneficial consequences associated with the impact of planned developments are derived from increased water
availability in the dry season, and reduced flood prevalence in the wet season. The reduction in flood risk due to the decline in the wet season flows and water levels would be a large socio-economic benefit of these development plans, potentially reducing the duration and extent of affected regions by more than 20%. In addition, increased dry season flow would greatly enhance agricultural productivity, enhance water security, and minimize conflicts between consumers. Environmental flow



could also be secured which may help some aspects of ecosystem productivity. Increases in water levels might also reduce
energy costs associated with water pumping, and better facilitate dry season navigation.

However, there are many negative consequences to the reduction in flood extent and duration associated with the
planned developments of 2020–2040. Hydropower projects in the Mekong are projected to trap considerable parts of the
sediments and the nutrients it contains in the reservoir behind the dam wall, reducing their transportation downstream and
subsequent distribution across the floodplain (Kondolf et al., 2018; Kummu et al., 2010; Schmitt et al., 2018; Schmitt et al.,
2017). The reduction in sediment transport rates associated with reduced wet season flows and sediment trapping upstream
inevitably leads to sediment-starved water flow downstream. This in turn leads to increased rates of channel incision and
accelerating riverbank erosion as river waters gain in-situ material for transportation up to carrying capacity (Morris, 2014;
Darby et al., 2013). The drop in soil fertility (nutrient bound to sediment) throughout the downstream floodplains would
result in a great challenge for ecosystem productivity (Arias et al., 2014), rice production (Boretti, 2020) and the
sustainability of flooded forests (rich habitats for fish and other species) (Arias et al., 2014). Dams also act as barriers
disturbing fish migration between upstream and downstream sections essential for feeding and breeding, resulting in
fisheries losses (Ziv et al., 2012). In addition, the increasing dry season water levels will disturb various river works - for
instance, the low water level condition is favourable to river channel maintenances (dredging) and constructions of water
infrastructure, usually started and very active during the dry season months.

Whilst higher economic damages from flood disasters are proportional to extended flooded areas, intensifying flood
depths, and prolonging flood durations, there are counteracting positive impacts associated with floods, including the
transport of nutrients and increased fisheries productivity. Increasing flood extents widen the coverage of fertile agricultural
land (Lamberts, 2008), which implies a more extensive production of rice - the most important agricultural activity in the
Cambodian Mekong floodplain. In contrast, a substantial reduction in flooded area would lead to a fall in flooded forest, a
rich habitat for fish and other species (Arias et al., 2014; Kummu and Sarkkula, 2008), leading to a decline in fisheries and
other ecosystem productivities. These benefits from an extended flood extent need to be balanced against the detrimental
impacts of deep flood depths and long flood durations, which can be catastrophic to crop yields across the floodplains.
Therefore, suitable flood conditions should be well determined for a better trade-off with the developmental impacts.

### 4.3 Limitations and perspectives for future research

Several studies have been conducted to understand hydrologic processes within the Mekong floodplains including parts of
Cambodia, Tonle Sap Lake Basin, and Mekong Delta. Different considerations have been taken into account for the analysis
in previous researches; they include but are not limited to (1) water infrastructure development, (2) climate change, (3) sea
level rise, (4) land use and land cover change, (5) population growth, and (6) climatic related phenomena. However, the
present study is targeted to gain insight into how the combination of water infrastructure development and climate change
will affect the Cambodian Mekong floodplain in terms of hydrological and flood patterns. Under climate change scenarios,
the future rainfall and temperature were assumed respectively to be wetter and warmer, while the land use change was



considered unchanged in the future. The effect of sea level rise and tides was also excluded in this study, but any tidal effects would have a minor influence on the water level fluctuations at hydrological stations in the Cambodian Mekong River (Dang et al., 2018). Another limitation of this study is our inclusion of just one GCM and one RCP. Whilst there is a large degree of variation between GCMs in the region, the general trends are consistent (wetter wet seasons, and drier dry seasons), and our choice of GCM represents the median magnitude of these directional changes (MRC, 2017). Nevertheless, the future inclusion of multiple GCMs and RCPs could lead to uncover (1) the lower/upper bounds or extreme events of projected climate, (2) dissimilar degrees of change and impact to different sectors, and (3) a plausible range of future change and impacts.

The impact of dam operations will be opposing those of irrigation, as they may lower hydrological conditions during the dry season, which are expected to be increased by the dam operations (Lauri et al., 2012). Therefore, to minimise uncertainties in terms of future directions and magnitudes of changes resulting from these key drivers, reliable and up-to-date data, and detailed information of key drivers should be well considered. Future research should employ finer resolution climate models, and more GCMs and CMIP-6 scenarios in combination with a small scale decision support tool set-up; as well as satellite-based image analysis to assist in evaluating a comprehensive study of the flood vulnerability or Water-Energy-Food Nexus in the Cambodian Mekong floodplain for the present and future conditions.

Another relevant research direction is the prediction of future land use and river morphological changes. This could generate a key input for a more realistic assessment of hydrological and flood alteration. River sand mining has been very active in the Cambodian Mekong River and its main tributaries as rapid and on-going urbanization requires a massive amount of sand, which is an important material not only for construction but also for backfill (Boretti, 2020; Hackney et al., 2020). River bank collapses, directly or indirectly associated with excessive sand extraction, have been very severe. Moreover, many floodplains and wetlands have been filled up by sand and transformed into urban areas, resulting in a critical change in river morphology and landscape along the river channels and throughout the floodplains. More importantly, these alterations are still being perpetuated without the full impact of their occurrence being understood or accounted for.

Floods are an essential component of the landscape for both the people and the ecosystem of the Mekong Basin, but they also pose significant hazards and losses when the magnitude is too great to handle effectively. As the development of water infrastructure could cause a decrease in flood conditions and climate change may reverse such impacts, it is still unknown what the desired flood water level and flood duration should be. This has led to a great difficulty in proposing optimum flood protection measures while maximizing dam benefits. Therefore, another potential research topic is the determination of the ideal flood conditions for a maximum productivity from both the agricultural and ecosystem perspectives.

The intended purpose of these future researches is to provide valuable information and assist governments, policymakers, and water resources engineers to foresee future threats of different intensities. Moreover, their results would be helpful in formulating better water resources management strategies, and in elevating all living things' resilience to the future challenges for the sustainability of resources within the floodplain.


## 5 Conclusions

By combining the effects of development activities and climate change, this research performs a cumulative impact assessment of the hydrological regime changes in the Cambodian Mekong River and flood condition alterations within its floodplains. We integrated the planned development activity of six central sectors throughout the Mekong River Basin: hydropower, irrigation, navigation, flood protection, agricultural land use, and water use. The study also attempts to isolate the individual impacts of climate change, mainstream dams in the Lower Mekong Basin, and hydropower mitigation investments. The modelling results show high sensitivity of hydrological and flood condition responses to the drivers considered as part of our analysis, highlighting the importance of properly characterising the directions and magnitudes of these changes. This study will contribute to the delivery of more precise information about the expected hydrology and flood behaviours resulting from future development activities and climate change, and assist in strategic plan formulation and decision making processes in the dynamic Mekong region.

The key results from this research demonstrate that the monthly, sub-seasonal and seasonal hydrological regimes in the Cambodian Mekong River will be subject to substantial alterations under the 2020 development scenario, and even larger alterations under the 2040 development scenario. Both development scenarios exhibit the same generic pattern of decreasing hydrological conditions during the early wet season months, whilst increasing water discharge and water levels in the dry season months. The degree of hydrological alteration under the full development scenario (2040) is counteracted by the effect of climate change, which is projected to intensify the onset of wet season months and exacerbate water deficiencies in the dry season months. The removal of mainstream dams along the Lower Mekong Basin and the implementation of hydropower mitigation investments also counteract the impact of the 2040 planned developments, diminishing the reduction in wet season flows and the increase in dry season flows across all regions. The planned 2040 developmental impact on flood characteristics is to significantly reduce extent, duration, and depth throughout all provinces, with the largest reductions being in Kampong Thom province of 10.5% for area and 14.1% for duration. Again, these reductions in flood characteristics are counteracted by both climate change and mitigation measures, in some provinces to such an extent that they display slight increases in flood extent, depth, and duration, most notably in Svay Rieng.

The positive and negative implications of developmental impacts on water availability, flow alterations, and particularly flood regime alterations should be carefully considered when determining the level of investment to place in counteracting measures. Reduced wet season flows and the associated reduction in flood extent, depth, and duration have demonstrable flood protection benefits that reduce the socio-economic impact of damage to infrastructure, crop yields and land, and hazards to public health, whereas increases in dry season flows have the benefit of increased water availability for irrigation, consumption, and maintaining environmental flow. However, there are negative consequences to the impacts of the planned 2040 development including a reduction in fisheries productivity, sediment trapping and a decline in nutrient supply to the floodplain, and a reduction in floodplain ecosystem productivity including flooded forests. Balancing these trade-offs will be an essential component of any successful floodplain management strategy put in place to address future climate change and



uncertainty in a sustainable manner. A timely preparedness will be essential to avoid future economic and environmental
damages, as well as safeguarding the wellbeing of vulnerable communities living throughout the Lower Mekong floodplains.

*Data availability.* The observed meteorological and hydrological data can be ordered from the Mekong River Commission's
Data Portal (https://portal.mrcmekong.org/home). Other supporting data are accessible through the associated references.

*Supplement.* The supplement related to this article is available online at: …

*Author contributions.* SH, AH, and MK jointly developed the idea and designed the details of this study. JK performed the
model construction. SC and SH carried out the model calibration and validation, and produced the figures. SH and AH
conducted the overall analysis with support from MK and PH. SH and PH prepared the manuscript with contributions from
AH and MK. The manuscript was edited by AH and MK.

*Competing interests.* The authors declare that they have no conflict of interest.

*Acknowledgements.* The study was funded by Academy of Finland funded project WASCO (grant no. 305471) and
additional funding was received from European Research Council (ERC) under the European Union's Horizon 2020
research and innovation programme (grant agreement No. 819202). Authors are also sincerely thankful to all relevant
organizations for supporting information and data to conduct this study.

*Review statement.* This paper was edited by … and reviewed by …

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
