# Peer review of "The Cambodian Mekong floodplain under future development plans and climate change"

_Natural Hazards and Earth System Sciences, 2021_

## Author Response (AR1)

We would like to thank both of the reviewers for supplying such thoughtful insights and comments on our article. In order to address these concerns, we have made major revisions to our original manuscript, completing a total re-analysis using a novel modelling setup and data supplied from Hoang et al (2019) and Triet et al (2020). We have now employed a different structure to our scenarios and added additional climate change components, utilising an ensemble of 5 GCMs for both RCP 4.5 and RCP 8.5 projections.

In addition to these broad alterations to our analysis, we have also addressed each of the specific comments that the reviewers have identified, the details of which we have given below.

Thank you again for your time and consideration, and we look forward to your decision.

**Response to reviewer 1 general comments:**

Comment 1.1: The article mixed the impact of Cambodian or Mekong floodplain and Mekong Delta. It does not seem clear which area refer to this or both are the same.

> Reply1.1: We have now made this distinction clear in the study area section of the methods (lines 112-117) and used this terminology throughout the article.
>
> *"The study area is located in the downstream part of the Cambodian Mekong River Basin (excluding the Tonle Sap Lake region), also known as the Cambodian Mekong floodplain. The area is about 27,760 km2 and extends along the Mekong mainstream from Kratie province to the Cambodia-Vietnam border."*

C1.2: Check the definition of flood season and wet season. Or refer to the same season?

> R1.2: We have now standardised our language to only refer to a wet and dry season.

C1.3: Check the definition of the wet and dry season – from which month to which month?

> R1.3: We have now added the following text in section 2.1 (study area)...
>
> *"The wet season runs from June to October, and the dry season runs from November to May."*
>
> Where we refer to months outside of these ranges, we have stated the months in parentheses afterwards.

C1.4: The research paper seems to miss the discussion of the results and propose solution and mitigation measure to overcome.

> R1.4: We have included a fairly comprehensive discussion of the results and their implications in the discussion section, which we have now revised to reflect the change in our re-analysis. We don't feel the scope of this paper is wide enough to talk about mitigation or solutions to the drivers we investigate (climate change, hydropower development, or irrigation).

C1.5: The baseline period (1985-2008) is a bit old - consider extending it to more recent years.

> R1.5: Thanks for raising the baseline period -issue. We reflected the used baseline on literature and basin development, and decided to actually use an older time period as the baseline (1971-2000) for our re-analysis. . We assert that this longer time period is better

suited to represent a time relatively free of hydrological alterations by large infrastructure and irrigations projects. Though there were a few dams constructed by the late 1960's, most of the Mekong basin remained relatively unaffected by large hydropower through to the late 1990's as it was hampered by the cold war conflict era (Soukhaphon et al., 2021). The ecosystem and people's livelihoods have become accustomed to the natural flood cycle conditions (i.e. pre-2000) over the centuries. Thus, it is justified to have that period as a baseline. Further, using this baseline of 1971-2000 also enables us to make direct comparisons with other studies that have used the same baseline in the region (Hoang et al., 2019; Triet et al., 2020).

The justification for the used baseline is now given in the revised manuscript (page 10, lines 198-201).

*"The use of 1971-2000 as our baseline represents well the hydrological state of the basin before major alterations were introduced (Soukhaphon et al., 2021). Including years after 2000 in our baseline would introduce significant hydrological and irrigation influences that would prohibit a thorough examination of these in isolation as part of our simulations."*

C1.6: It is not clear how the selection of climate change dataset to apply in this study. This would lead to uncertainty of result analysis and interpretation.

R1.6: We agree that this was not clear. In the revised analysis, we now use an ensemble of 5 GCMs that have been shown to be applicable in this region.

C1.7: An accurate description of general areas/places is sometimes confusing. This happens in many places throughout the article. Particular attention should be paid when revising the article.

R1.7: As part of the re-analysis we have now updated our description of the results and tried to pay attention to reducing this confusion.

*Specific comments of reviewer 1*

C1.8: Line 93: Add "Hydrological" condition...

R1.8: We have now added 'Hydrological' (line 121).

C1.9: Line 97: Remove "extreme"

R1.9: We have now removed 'extreme' (line 129).

C1.10: Line 99: "catchment" annual average temperature refers to the catchment of the Model or study area? Rewrite this sentence to make it clearer.

R1.10: We have now clarified that this refers to the study area (lines 130-134).

*"...the annual average temperature across the study area varies from 26.9°C to 28.2°C, with mean monthly temperatures between 30°C during the hottest months (April and/or May), and 26°C in the coldest month (January)."*

C1.11: Line 102&103: What is the wettest month and driest month?

R1.11: We have now added this data (line 135).

C1.12: Table 1: Add year of Topography.

R1.12: We have now added the year.

C1.13: Table 1: What is Geospatial data? Year?

R1.13: This is no longer applicable with the re-analysis, so we removed it from the table.

C1.14: Line 135-139: These indices are not common - therefore good to describe more comprehensively.

R1.14: We have now used precision and recall indices to assess the performance of the flood extent model, which we detail in the text (lines 214-218).

*"Recall evaluates what proportion (0-1) of the flood derived from remote sensing images are identified by the simulation. Precision evaluates what proportion of the simulated extent agrees with the remote sensing. "*

C1.15: Line 162-188: This section can be summarised with a table. This would improve the readability of the section. Hence, it is not sure if Table 2 is really useful.

R1.15: We have re-written this section to describe the scenarios in our re-analysis.

C1.16: Line 226-252: It is hard to read this section. Please consider rewriting it.

R1.16: We have re-written this section to describe the scenarios in our re-analysis.

C1.17: Table 3: This table may need to reconceptualise for better information visualisation.

R1.17: we changed the formatting of the table; hopefully it reads better now.

C1.18: Line 253-294: Finding in Section 3.3 looks really interesting – however a simpler presentation of the findings would help improve readability and convey key messages.

R1.18: We have now revised the entire section as well as simplified the figure to boxplot representations.

C1.19: Line 295-316: Not sure if the description of Cambodian provinces in this section is important for the international context of this journal.

R1.19: Whilst we acknowledge that this is a valid concern, we have included the provincial analysis to highlight the complexity and heterogeneity of the region, which further shows that this type of localised floodplain study is necessary in addition to the large scale basin-wide undertakings.

C1.20: Line 316: It is more appropriate to mention that your model outperformed others – but it does not mean your model is more accurate than others. Consider revising this sentence. This also applies to other parts of this section.

R1.20: This is a very good point and we have now refrained from discussing accuracy and instead compare the relevant model performance metrics (lines 390-399).

*"The model performance metrics achieved by our hydrological simulation of water discharge and water level for the baseline period of 1971–2000 at all four monitoring stations (Kratie, Kampong Cham, Chroy Changvar and Neak Loeung) exceed existing studies within the same region…"*

C1.21: Line332: Uncertainty may cause by the climate change dataset used in this study – but it may not come from the simulation of climate change of this study. Consider rephrasing this sentence.

R1.21: We have re-written this section to reflect our re-analysis.

C1.22: Line 425-426: Water-Energy-Food Nexus could not just look at the Cambodian Mekong floodplain alone. Consider revising this sentence.

> R1.22: We have amended this to reflect the wider reach of the Water-Energy-Food Nexus (lines 498-502).

**Response to reviewer 2 general comments:**

Comment 2.1: The study is very ambitious in that it considers a large range of factors, sectors and drivers of change; also, the methodology encompasses a large set of advanced modeling tools.

C2.2: The title, abstract and motivation of the paper feature climate change prominently, along only another driver of the changes analysed. Still, the way climate change is treated in the study framework is less than optimal. Regarding the climate model used to simulate future changes in climate: one single model was used, which doesn't allow to address the relative uncertainty; the model chosen is dated (CMIP5 generation); it is not clear that the use of its results is validated through comparison with observations; GCMs are generally considered inadequate for to study hydrological processes at such fine sale over a small domain, where Regional Climate Models are more appropriate and overcome mostly shortcomings that are not negligible when looking at precipitation extremes in a monsoonal climate.

> R2.2: We agree that this was the main shortcoming of our original analysis, and have undertaken a complete re-analysis in order to address this concern. Our re-analysis now includes an ensemble of 5 GCMs and two RCP levels that have a precedent for use and suitability in the region. The chosen 5 GCMs were selected as they performed best in the region (Hoang et al. 2019).
>
> Whilst we acknowledge that the models we use are from the dated CMIP5 generation, we now justify their use. We have included an analysis in our supplementary (Table S1) that compares the mean precipitation and temperature for the wet and dry season across our study area between an ensemble of six CMIP5 models and the equivalent models from the CMIP6 generation and show that the differences are very small.
>
> The use of newer CMIP6 generation models would require us to fully rebuild the basin-wide scenarios, which is beyond the scope of the work. And as the difference between the CMIP5 and CMIP6 scenarios is very small, it would not change the main findings nor our conclusions. Further, by using the existing basin-side scenarios, allows us to compare our results to basin-wide studies as well as studies done for Vietnamese part of the delta. Therefore, we chose to use the CMIP5 scenarios for this study but, as stated above, we now use an ensemble of GCMs.

C2.3: It seems that the paper presents the results of an advanced framework that integrates multiple types of models and uses a large variety of datasets. It is not possible to me to judge, however, whether the setup is appropriate, due to a lacking explanation of the experimental setup. In particular, it is hard to understand how each model in the set of those adopted relates to each other.

> R2.3: We agree that the explanation was inadequate; apologies. We fully revised the model set-up and procedure for the updated analyses. In short, we use now a combination of three models: Mekong basin-wide distributed hydrological model (IWRM-VMmod,) whole Mekong delta 1-D flood propagation model MIKE-11 (boundary conditions from IWRM-VMod and sea level in South China sea, including detail modelling of the Vietnamese part of the delta with its canals and sluice gate) and 2-D flood duration and extent model IWRM-Sub for Cambodian Mekong floodplains, that enables detail floodplain modelling.

Also, we have taken extra care in ensuring a more thorough explanation of how each component adds to the end result (revised Section 2.2), including a conceptual diagram of the model system that we have included in the main text (Fig. 2).

C2.4: The setup of scenarios and their explanation are lacking. Mainly: the use of one single future climate scenario, a practice that is strongly discouraged in the field; and the lack of a plan to understand the effects of climate change on each of the scenarios of infrastructure development. Further, scenarios of socio-economic development (their present IPCC iteration being the Shared-Socioeconomic Pathways, SSPs) seem to matter in the analysis included, for what concerns land use and agriculture, water use, irrigation etc.

R2.4: We agree with the reviewer that the description of the scenarios was not clear enough. Having now changed the entire model setup, it also allowed us to update the scenario setting. We have given a more comprehensive description of each scenario, as well as included references to more details from the original sources (Section 2.4).

C2.5: A notoriously biased and inaccurate elevation dataset is used for the flood modeling, where improved datasets exists that are even included in other parts of the work.

R2.5: We agree that there is a mismatch between the datasets. The flood extent and duration model IWRM-Sub we used, is based on older elevation model and combined with detail bathymetry survey data. Due to how the model is constructed, partly because of the combination of SRTM and detail bathymetry data, unfortunately we cannot change the underlying DEM. As we use these 90 m resolution datasets at the aggregated 1 km resolution, any differences are unlikely to impact our results dramatically.

C2.6: Whereas the concept of transboundary water management has gradually gain firm footing in the last years and decades, this study stands in stark contrast with such universally preferred practice in that the situation downstream of the national boundary is neglected. It seems reasonable that focusing on one country, Cambodia, enables a more detailed and focused analysis, allows to neglect the effects of coastal processes and sea level rise, and may be also justified on grounds of dataset availability; but in the context of the lower reaches of the Mekong river it seems arbitrary to cut the modeling and analysis domain at the boundary with Vietnam. I do not ask the authors to repeat their analysis on a larger domain, but I suggest that this aspect should receive (concise) attention, and that the implications of the study, including any policy recommendations the authors may choose to draw, reflect recognition of this limitation. It would indeed be unwise to recommend policy based on knowledge of effects for only one of the countries in the lower Mekong, before quantifying the effects onto other territories downstream.

R2.6: This is another very good point, and one that we have tried to address in our re-analysis. With our novel setup of three models (see our reply R2.3) we are now able to take into account the sea level rise as well as the complex setup of canals and sluice gates in Vietnamese Mekong Delta. This was done by using boundary conditions to the Cambodian part model (IWRM-Sub) from MIKE 11 model (see Fig 2). MIKE 11 is set-up for the area by Triet et al. (2020) who focuses on the impact of climate change on the Vietnamese Mekong delta. Through their model application to the entire delta, the downstream effects of coastal processes and sea level rise are implicitly incorporated into our re-analysis, though we have not enlarged our focused study area nor made inferences about the impacts of our results on the wider delta region.

*Specific comments of reviewer 2*

Abstract

C2.7: I suggest that the first three sentences could be condensed so that the abstract can soon reach the core of the paper at hand.

> R2.7: We have now condensed the first three sentences as suggested and it reads much better.

C2.8: When in the fourth sentence you mention impacts, it is not clear what these impacts refer to: which is the impacting phenomenon? I suggest you take the occasion to explain that you are scientifically assessing the implications of planned interventions in a context of changing climate (or some similar formulation).

> R2.8: We have done as you suggested and rephrased this section (lines 6-9).

C2.9: In the following sentence you mention modeling, but you have not explained what type of modeling: please use some words to lay out the methods of the paper.

> R2.9: We have now added this (page 2, lines 11-12):
>
> '*distributed hydrological (IWRM-Vmod) and flood propagation (MIKE-11 and IWRM-SUB) modelling analysis*'.

C2.10: Following sentence: isn't there overlap between 'monthly' and 'sub-seasonal'?

> R2.10: We agree with this comment, and now just say monthly and seasonal (lines 14-15).

C2.11: How come a 'scenario', i.e., a formulation of the future (as implied also by the use of the future tense), refers to year 2020, which is in the past?

> R2.11: We have now revised our scenarios and no longer have a 2020 scenario.

C2.12: What concretely is altered, in the 'hydrological regimes': is it discharge?

> R2.12: We have now clarified that this refers to 'regimes (discharges, water levels, and flood dynamics)' (line 15).

C2.13: What is 'hydropower mitigation investment'?

> R2.13: This is no longer part of our scenario structure.

C2.14: I think the last sentence of the abstract is vague and not connect to the results of the study.

> R2.14: We have now amended the last sentence to better reflect our findings (lines 25-27)
>
> '*Our findings highlight the hydrological complexity and heterogeneity of this region, and demonstrate the substantial changes that planned infrastructural development will have on these ecologically fragile floodplains.*'.

Introduction

C2.15: The part on the benefits of annual (seasonal?) flooding is very important, and it's essential that it be well explained since the notion and key concept don't receive much focus in the literature. I suggest adding a sentence to explain how annual flooding improves water availability in the dry season, as this is not obvious.

> R2.15: We have now expanded this section to explain about aquifer recharge (lines 45-47).

> *"...floodwaters play an important role in the recharging of aquifers and ensuring the hydrological connectivity of the floodplain, which is essential to maintaining ground water resources for use during the dry season…"*

C2.16: Please, make reference to Fig. 1 when pointing to locations in the study area, so the reader can better follow the explanation.

> R2.16: We now refer to fig. 1 in several parts during the introduction.

C2.17: Line 33: substitute comma for semicolon.

> R2.17: We have done this (line 55).

C2.18: L 39: please rephrase sentence starting with 'moreover' (add a verb). Also, explain what you mean by 'energy' here.

> R2.18: We have changed the beginning to 'In addition' and have removed mention of energy. (lines 54-57).

C2.19: L 41: 'Hydropower' doesn't seem a phenomenon of concept that can impact the water cycle: I think you should rather talk of 'hydropower dams/reservoirs/infrastructure' here.

> R2.19: We added infrastructure (line 53).

C2.20: L 41: you mention future projections here, but don't add any detail about e.g. which climate scenario and which future time horizon these refer to: please explain better. Also, which study do these come from? I suspect they don't come from both of the studies mentioned. In the following sentences these aspects are treated more systematically, but please try to be more specific also there, e.g. in terms of scenarios and time horizons.

> R2.20: We have now clarified the future projections in question for each of the examples given in the text (lines 57-74).

C2.21: L 55: you mention 'in opposition to climate change', but the reader should know what is the expected effect of climate change on the metrics treated here.

> R2.21: We have removed this as it is indeed ambiguous, and the precise impact of climate change hasn't yet been clarified to the reader.

C2.22: L 59 on: is 'peak inundation area' the same as 'flood extent'? if so, please use the same term to avoid confusing the reader. Also, you express change in terms of percentage and also in terms of surface area, which prevents understanding the difference between such changes.

> R2.22: We have now changed this to 'flood extent' (line 81).

C2.23: Please check that it is clear to the reader how this study goes beyond the one of Try et al. 2020a. I understand that that study's limitation is that infrastructure development was not considered?

> R2.23: We have now stated that Try et al., 2020a considered only climate change in isolation, and at a different time scale (2075-2099) (lines 76-77).

C2.24: L 80: I'd leave 'important' out from this sentence, as it is a subjective judgement. Also, in the following lines, I'd avoid mentioning 'global sustainable development goals' as it seems to only serve to aggrandize the study. But this is only my opinion, please judge for yourself.

> R2.24: We have removed 'important' and have changed the last sentence to read '*The results of this study may contribute to formulating adaptation and mitigation strategies to flood-*

*prone areas that balance the need for flood prevention and water resource allocation against the ecological functioning of the floodplain.*' (lines 108-110).

C2.25: Please keep abbreviations to the minimum necessary. E.g. UMB and MRB are almost never used.

R2.25: We agree that these are redundant and have replaced them.

C2.26: Is it warranted to dedicate virtually all attention in the introduction to the problem of flooding, and not e.g. to the problem of scarcity of water for agriculture?

R2.26: Whilst we agree that the issue of water scarcity is an important one to the region, and for that reason we have discussed it briefly in both the introduction and the discussion sections, the focus of this analysis is very much on the flooding aspects of these expected hydrological changes. Therefore we feel justified in making flooding the main topic of the introduction.

Materials and methods

C2.27: Explain MRC. Don't use abbreviations HR, TR and NE.

R2.27: We have explained MRC and changed the metrics we use for the flood extent verification, so no longer use the abbreviations HR, TR.

C2.28: It is a bit disappointing that the elevation model used is the SRTM 90 m resolution one, whose vertical inaccuracies are known. Was it not possible to access lidar data for this relatively small area, or datasets that improve on SRTM, e.g. MERIT (Yamazaki et al. 2917) which you do use in the remote sensing part of this work. Please mention this in the methods. Also, the land-use map is quite dated (is it from 2003): can you justify the (necessary?) choice?

R2.28: As we replied to Comment C2.5 , the DEM is an embedded component of the model that is combined with the survey based bathymetric data, which wouldn't be possible to separate at this stage. However, as we amalgamate the 90 m SRTM data to a 1 km scale, the difference between the updated MERIT dataset would be negligible.

C2.29: L 109 on: Please explain more clearly how each model stands in relation to the other, and what each model uses as input and what output is analysed and further used. There is mention of hydrological model IWRM-VMod, floodplain (hydrodynamic?) model IWRM-Sub, SWAT, IQQM and ISIS. Unlike stated, Table 1 doesn't describe models. Fig. S1 does a far better job at this, but the key points of the methodology should be clear without opening the supplementary material.

R2.29: As we now changed the entire modelling structure, we have included much more description of the models used, how they link together, and the driving data that is required for each, all of which is now summarised in Fig 2 as well as within the main text (Section 2.2. *Modelling structure and datasets*).

C2.30: Table 1: what is 'climate change' here?

R2.30: This has been altered to 'climate change projections of temperature and precipitation'.

C2.31: L 130: can you please explain the choice of these four stations: are these the only available? It would have seemed reasonable to have selected also a station in the tributary and distributary towards Tonle Sap, due to the complex and seasonal behavior of this river trait.

R2.31: We did not select a station along the Tonle Sap for comparison within the small scale floodplain model as the Tonle Sap discharges are a boundary condition from MIKE 11, fed to the hydrodynamic model and so are not simulated in the IWRM-Sub model.

C2.32: L 132: 'For the range adopted for performance rating see ASABE (2017).' Please provide further explanation for this. E.g., what 'range'?

R2.32: We have removed this reference as it was unclear.

C2.33: L 138: please explain better what NE is.

R2.33: We no longer report NE, instead we report recall, precision, and the ratio of the flooded extents.

C2.34: L 149 on: I cannot understand the explanation of the satellite-based images. See following points:

What are these stacks composed of, daily flood maps? Please clarify the explanation of the percentile maps: what percentiles did you take, what do they represent? 'permanent' and 'temporary' is not clear, do you mean permanent water bodies and flood waters? Why 'default' values? What is the water index? What threshold values? If the explanation is too technical for the non-expert in remote sensing (like me) to follow, please provide a simplified, though understandable, broad explanation in the main, and add technical, though clear, details in the supplementary. Also, please add necessary explanation in the caption of fig. S2, for the abbreviations and each step in the data processing.

R2.34: We have reduced and clarified the description of the flood extent map generation (lines 206-212) and clarified the abbreviations used in the Figure S2 (now S1).

"*The SWMT is a Google Appspot based online application developed by Donchyts et al. (2016). A stack of Landsat (4 and 5) data was generated using SWMT from 1984 - 2000. This stack of images was then used to generate a water index map using the Modified Normalized Difference Water Index (MNDWI) (Xu, 2006) to distinguish between water and non-water areas, which were then adjusted to account for dark vegetation and hill shadows using a Height Above Nearest Drainage (HAND) map (Rennó et al., 2008).*"

C2.35: L 165: Please re-think the explanation of the scenarios to see if you can make it more straightforward. Also, please explain the overall thinking behind the formulation of the scenarios: what overall questions are you trying to address with such study setup? You explain some of this in the Results, e.g., at lines 248 on, but it would seem necessary to explain this in the Methods. Some more specific points follow. What do you mean by 'definite'? It is highly confusing to the reader that you define year 2020 as a future scenario. I cannot see the reasons behind this choice: please either provide clear reasons or modify the definitions. Why didn't you add the effect of climate change to all scenarios of future infrastructure? Or, even better, why didn't you plan to simulate all future infrastructure scenarios both with and without climate change? I understand this would multiply work and results and complicate their presentation, but please discuss whether this is a warranted simplification. Are dams of the central panel in fig. S3 already realized, or are these 'plans'?

R2.35: Our reanalysis now has a different scenario setup which we have been careful to go into much more detail about (see Section 2.4 and Table 2), and have referred to the original scenario formulations where appropriate.

C2.36: L 173: there is no Pla2020 scenario in table 2.

R2.36: There is no longer a Pla2020 scenario.

C2.37: L 174: are LMB 2020 dams only two, in addition to those of 2007 in scenario BASE? If so, don't mention 'Xayaburi and Don Sahong only', or it will seem like an arbitrary choice to include only two. Further, when describing e.g. scenario Pla2020 just outline the differences from the BASE scenario, without mentioning everything that is included again. Also, clarify how factors like agriculture, land use irrigation change across years 2007, 2020, 2040: what are the sources of these datasets and what drivers and socio-economic scenarios do they presuppose? How are these factors included in the modeling, how are they parametrized?

> R2.37: These comments have been addressed by changing our definition of the scenarios, which now clearly distinguish between baseline and future hydropower and irrigation expansion in isolation as well as in combination.

C2.38: L 179: 'IPSL-CM5A-MR under RCP4.5' this requires explanation. Define (and cite?) IPCC, GCM and RCP4.5. I don't think that one model can 'represents the range of uncertainty inherent in the GCM climate change projections'. Do you mean that its results are representative of the IPCC ensemble of GCMs because they fall around the mean/median of the ensemble ones? What do you mean by 'covers monsoon seasonality': does the model successfully capture the seasonal variations in precipitation that characterize the local summer monsoon? Did you only use the results of the GCM for the scenario(s) of year 2040, or did you use them to simulate the situation at year 2007 and 2020, to assess biases and differences with the same simulations forced with observed temperature and rainfall? Was the output of the model bias corrected? What is the reference for this GCM's setup?

> R2.38: We have now changed our future climate projections used in the analysis to address these concerns. We now use an ensemble of 5 GCMs and 2 RCP levels that have been shown to be suitable for use in this region. See more at our reply R2.2.

C2.39: L 184: what does 'mitigation' here refer to? What do these plans try to mitigate?

> R2.39: We no longer use this term here are our description of the scenarios has now changed.

C2.40: Some questions I believe where not addressed: How long were the simulations of scenarios? What are the past meteorological data based on? What spatial resolution? How where the precipitation data from the GCM downscaled? How is the effect of dams/reservoirs/hydropower stations included in the simulations? What dam operation decisions and principles have been included, what is the level of confidence about these?

> R2.40: These concerns have all been addressed in the formulation of our new scenarios, with a brief description of how each represents a future development component given in the text along with a more thorough description given in the original source of the scenarios, which has been referenced extensively in the main text.

Results

C2.41: Table 3: please explain abbreviations.

> R2.41: We have now expanded the abbreviations in the caption.

C2.42: L 200: stations cannot overestimate: the model either over- or underestimates.

> R2.42: We have re-written most of this section to reflect the results of our new scenario formulations.

C2.43: L 206: what floods are presented and discussed here? Because this was not explained in the methods, the reader is not sure that this is the maximum flood extent in the 1985-2008 period, or a specific return period, or something else.

R2.43: These scenarios and what is being reported are now better described to make this clear (lines 338-343)

"*We compared year to year the impact of each development scenario against the S1_baseline (1971-2000) on the total flooded area across the study area (Fig. 5). Scenarios S2-S4 use the same driving climate data as the baseline scenario (S1), and so the variability in the impact shown is significantly reduced to produce consistent impacts for all years. Whereas scenarios S5-S12 are driven by future climate data projections, so that the variability in comparing year to year is significant.*"

C2.44: L 208: please remind the reader that the SWMT data represent the (proxy for) flood observations. Also in the figure, please from which models and datasets the images come from.

R2.44: We have re-written this section and the figure as the results have changed significantly (lines 265-272 and Fig. 3).

C2.45: L 209: it is indeed interesting that the overestimation could be (partly) an artifact in the strong difference in the resolution of the two datasets. Is there a way to test this, potentially by aggregating the higher-resolution data to a coarser resolution (in different ways) and looking at how the comparison then looks? The degree of agreement between observed and modeled floods is in any case remarkable.

R2.45: In our re-analysis this no longer seems to be the case. The modelled extent areas now match very well indeed in terms of magnitude, but there is still some discrepancy between the observed and simulated extents in some regions, though the overall coverage is a slight improvement from the previous analysis.

C2.46: L 210: what do you mean by 'scattering in the flood extent', and how would this affect the comparison with the modeled extent?

R2.46: We no longer mention this as it is not relevant any more.

C2.47: L 2020: I am not sure 'flow duration curves' are what is displayed in fig. S6, where the discharge is plotted versus exceedance probability. These are indeed often called 'exceedance probability' curves.

R2.47: We no longer include these as they were indeed vague and unnecessary.

C2.48: L 222: fig 3 is an effective way to summarize differences. But are these percentage changes with respect to the BASE scenario? Please explain.

R2.48: Yes, these are indeed percentage changes with respect to the baseline, which we have made much clearer both in the text, and the figure (Section 3.2 and fig. 4).

C2.49: L 235: this is not 'independent of climate variability'. It shows the impacts of planned developments if anthropogenic climate change were not to occur. Climate variability is a different concept. On another note, with respect to the GCM results: could you briefly specify somewhere how temperature and precipitation change in the simulated future climate with respect to the 1985-2008 reference period?

R2.49: This no longer appears in the text as it refers to scenarios that are not used any more.

C2.50: L 237: 'severely impact the hydrological functioning of the Mekong main channel' seems unwarranted wording. You should explain what 'function' the main channel has, and why the

modeled homogenization of the seasonal flows should be seen as a 'severe impact'. Maybe these sort of reasoning (also present in other parts of the Results) should be moved to the Discussion, where additional explanations and concepts can be added?

> R2.50: We appreciate that this sort of reasoning would be better suited to the discussion, and whilst it no longer appears in the manuscript, we have taken this comment on board and tried to ensure that we just present the results in this section.

C2.51: L 251: I suggest to leave judgement of what is the most sustainable course of action for decision making to the Discussion or other section. Results are not the place to add these. Why would it be preferable to minimize hydrological alterations? This requires arguing that doesn't fit here.

> R2.51: This section has now been completely re-written, but we agree that such discussion does not belong in this section.

C2.52: L 255: sentence staring with 'The comparisons' is unclear, please rephrase.

> R2.52: This no longer appears in the text due to the section being completely re-written.

C2.53: L 257: what does this range refer to , annual maximum flood values across the period?

> R2.53: This no longer appears in the text due to the section being completely re-written.

C2.54: L 260: readers will be familiar with flood being a peril and a problem that needs mitigating and reducing, so wording flood reduction as 'exacerbated' will at minimum read odd to many. I understand that in this context floods have both advantages and disadvantages, so I suggest a solution would be to exclude wording that expresses a value judgment and use more neutral and descriptive terms.

> R2.54: This no longer appears in the text due to the section being completely re-written.

C2.55: L 425: why is the 'water-energy-food nexus' mentioned here, with no motivation: what would be the merit of addressing that concept and how would that be possible?

> R2.55: This no longer appears in the text due to the section being completely re-written.

C2.56: In general, similar to my recommendation for Section 4.2, I suggest for Section 4.3 to stay closer to the core topics of this study, and avoid embarking in the longest possible list of things interesting about the Cambodian Mekong.

> R2.56: Although many of the comments in this section of the reviewers response were not directly applicable in the new draft, we have taken on board comments such as this one and tried to re-write the manuscript with this in mind.

Discussion

C2.57: I suggest to re-think the opening paragraphs to start-off the Discussion less as a listing of ways in which the present study is superior to previous comparable efforts.

> R2.57: Whilst we appreciate the reviewers thoughts, we feel it important to establish the credibility of our model simulations before we go on to report our results as something that the reader should have confidence in. However, we have reduced this section so that we can more quickly progress to discussing the results.

C2.58: L 319: I do not think here much can be said about confidence on future projections, especially not mentioning the 'error inherent within the GCM predations of future climate change', as only one climate scenario was simulated by only one model.

R2.58: We have removed this text from the draft. Further, as stated above we now use an ensemble of GCMs.

C2.59: L 323: how have you compared the performance of your setup for a single event to the performance of Fujii et al, 2003?

R2.59: We now realise that this is unclear in the text as we meant to imply that our multi-year analysis may not have as good a performance metric as the single event analysis carried out by Fujii et al., 2003. We have clarified this in the new draft (lines 405-407).

"*Whilst there are studies of flood extent within our study area that only focus on a single event rather than a multi-year analysis that slightly surpass our own in terms of performance metrics (Fujii et al., 2003)...*"

C2.60: L 326: 'confidence in our future projections of the Cambodian Mekong floodplain's response to changes in the flood hydrograph.' Please rephrase at it's not clear what response to what you are treating here.

R2.60: We have now changed this to read... 'confidence in our future projections of the Cambodian Mekong floodplain's hydrological response to planned infrastructural development and future climate changes' (lines 410-412).

C2.61: For the discussion of the implications of the changes in hydrological conditions and flood occurrence and extent, I suggest to stay relatively close to the results of this study. It is interesting to mention the cascading effects of more or less floods on different sectors and aspects of the socio-economy and the environment, but these seem to receive disproportionate space when considering that none of these consequences have been modeled in this study. Section 4.2 is largely a review of the literature of the ramified impacts of hydrological and flood changes, with limited connection to the study at hand.

R2.61: We appreciate the reviewer's insight and have now reduced this section to focus more on a discussion of the issues directly relevant to our modelling results.

C2.62: L 411: 'land use change was considered unchanged in the future': this contrast with the Table above that specifies that different land use is used to simulate years 2020 and 2040.

R2.62: We do indeed vary land use in the future projections as irrigation and agricultural expansion is now a component of many of the scenarios, which is better explained in the methods section.

Conclusions

C2.63: There is in general a large degree of overlap between concepts included here and in the Discussion and also in the Results. Please review each of these sections and try to reduce repetition to the minimum, repeating only the main concepts that you consider essential for the reasoning in each section.

R2.63: We agree that there was some repetition in the conclusions. We aimed to remove this repetition when revising the manuscript.

C2.64: Fig. 1

R2.64: Thanks for the good remarks; see replies below.

C2.64_1: Why in the small map some areas are dark and soma are light green?

R2.64_1: Cambodia is shown as light green in the map.

C2.64_2: Since country boundaries are important in defining the study area, I suggest making more clear from the map where each country lies.

R2.64_2: We now added the country border between Cambodia and Vietnam to the large map; this indeed makes the map more readable.

C2.64_3: The Tonle Sap lake is mentioned many times in the paper but the name doesn't feature in the map, please specify. Also, other names are mentioned in the text that are not reported in the figure, like the tributaries. I suggest to only mention names of places that are functional to understand the key aspects of the study, and to report those in the map for the many readers that will be unfamiliar with the area.

R2.64_3: We now added Tonle Sap as well as the river names. We aimed to make sure that only the relevant place names, such as provinces and station names, were mapped.

**References**

Shabeh ul Hasson, Salvatore Pascale, Valerio Lucarini, Jürgen Böhner. (2016). Seasonal cycle of precipitation over major river basins in South and Southeast Asia: A review of the CMIP5 climate models data for present climate and future climate projections. *Atmospheric Research, 180, 42-63.*

Hoang, L. P., Lauri, H., Kummu, M., Koponen, J., van Vliet, M. T. H., Supit, I., Leemans, R., Kabat, P., and Ludwig, F. (2016). Mekong River flow and hydrological extremes under climate change. *Hydrol. Earth Syst. Sci., 20, 3027–3041.*

Long P. Hoang, Michelle T.H. van Vliet, Matti Kummu, Hannu Lauri, Jorma Koponen, Iwan Supit, Rik Leemans, Pavel Kabat, Fulco Ludwig. (2019). The Mekong's future flows under multiple drivers: How climate change, hydropower developments and irrigation expansions drive hydrological changes. *Science of The Total Environment, 649, 601-609.*

Nguyen Van Khanh Triet, Nguyen Viet Dung, Long Phi Hoang, Nguyen Le Duy, Dung Duc Tran, Tran Tuan Anh, Matti Kummu, Bruno Merz, Heiko Apel. (2020). Future projections of flood dynamics in the Vietnamese Mekong Delta. *Science of The Total Environment, Volume 742, 140596.*

Soukhaphon, A., Baird, I. G., & Hogan, Z. S., 2021. The Impacts of Hydropower Dams in the Mekong River Basin: A Review. Water, 13(3), 265. https://doi.org/10.3390/w13030265

---

## Author Response (AR3)

We would like to sincerely thank the reviewer for their kind words and for once again providing such detailed and thoughtful comments; these have highly improved our manuscript. Below you will find responses to each of the reviewer's comments outlining how we have addressed them in the revised manuscript.

**Response to reviewers' comments**

**Reviewer 1**

The manuscript has been substantially revised and the modelling structure and integration of various models were changed.

Overall, it is hard to closely examine the results and discussion in the paper as I strongly feel that the central questions to interpretation of the results and analysis rely on the proposed methodology and input datasets. Please see my general comments/suggestions below:

> Reply: Many thanks for your comments and suggestions on the revised version of the manuscript. We have now clarified the issues raised by the reviewer, as detailed below in our replies.

1. It is difficult to judge or verify the results from this large model integration as errors from one model could propagate to another, different time-steps used in hydrological (daily for V-Mod) and hydrodynamic model (sub-daily for MIKE11 and IWRM-Sub model).

> Reply 1: we do understand the reviewer's concern. However, while the validation results of the hydrological model (IWRM-VMod) and flood propagation model (MIKE 11) are presented in the cited documents (Triet et al 2017, Hoang et al 2019), we were able to validate the results also at the last step, i.e. the results from flood extent and duration model (IWRM-Sub). All the three model validations, after each step of the modelling setup, show that simulated results agree well with the observed ones.
>
> VMod: the NSE at the boundary to MIKE 11 is 0.68, i.e. high agreement with observed discharge (see Table 2 in Hoang et al 2019).
>
> MIKE 11: the NSE in Cambodian floodplain stations varies from 0.84 to 0.98, i.e. very high agreement with the observed water levels (see Table 1 in Triet et al 2017)
>
> IWRM-Sub: the NSE in Cambodian floodplain stations (excluding the upper boundary Kratie, that is not within the floodplain) for discharge varies from 0.80 to 0.81 and for water level from 0.85 to 0.87, i.e. very high agreement with the observed discharge and water levels, respectively (see our Table 3). For Kratie the NSE for discharge is 0.79 and water level 0.69, equally very high or high agreement with observations.
>
> We thus believe that the propagation of errors is not an issue in our article. We now communicate this better in the text by adding a comment on the possibility of compounding errors (lines 288-290).
>
> *"Using multiple models in succession can have the negative effect of compounding errors, however these results demonstrate that this has not unduly impacted our methodology as our estimations closely match the observations of flood extent."*

2. A workflow of modelling structure should be considered to add to this manuscript. This would be a better way to explain the modelling approach of this paper.

> Reply 2: the schematic illustration of the modelling setup is given in Fig 2 of the manuscript. We believe that this gives enough information about our modelling approach.

3. There is no explicit mention of the selection of GCMs, climate change datasets, and no detailed description of bias-correction or downscaling for the Mekong. What is the suitability of the selected climate change datasets for applying to this study?

> Reply 3: thank you for pointing this out; indeed, we failed to communicate this properly. The bias-correction and downscaling was done in Hoang et al (2019). This is explained in detail in Hoang et al (2016; 2019).
>
> The GCMs (ACCESS-1.0 (ACCESS); CCSM4 (CCSM); CSIRO-Mk3.6.0 (CSIRO); HadGEM2-ES (HadGEM); and MPI-ESM-LR (MPI)) were selected based on their performance regarding historic temperature reproduction (Huang et al., 2014), seasonal precipitation (Hasson et al., 2016) and climate extremes (Sillmann et al., 2013). The GCM data were downscaled using bilinear interpolation to a $0.5° × 0.5°$ spatial resolution and statistically bias corrected through quantile mapping method, following Piani et al. (2010). The adopted techniques for climate data downscaling and bias correction represent standard approaches for hydrological impact assessment studies such as van Vliet et al. (2013) and Yan et al. (2015).
>
> We now summarise this briefly in the revised manuscript (lines 237-240):
>
> *"These GCMs were selected based on their performance in reproducing historic temperature, seasonal precipitation, and climate extremes in the Mekong region. The GCM data were downscaled using bilinear interpolation and statistically bias corrected using a quantile mapping method. For full details see Hoang et al (2016; 2019)"*

4. Irrigation scenario is based on global projected expansion. It is advised to briefly describe if this global dataset and its variation were applicable for the Mekong.

> Reply 4: again, good request for clarification. For the baseline irrigation, data from the MIRCA - "Global Dataset of Monthly Irrigated and Rain-fed Crop Areas around the Year 2000" (Portmann et al., 2010) – was used. The MIRCA data set provides data on irrigated area and cropping calendar for 26 different crops at roughly 10 x 10 km resolution. This data was resampled to VMod resolution (5 km × 5 km). Since irrigated rice is the most dominant crop in the Mekong basin (accounting for over 80% of the total irrigated land) the focus in the scenarios was on irrigated rice. Although this is a global dataset, the figure pertaining to irrigated rice that we use refers to the Mekong region specifically as that is where the vast majority of irrigated rice is cultivated. Therefore it is indeed valid for use in this case.

5. One of the biggest challenges in modelling flow for the Mekong is a representative of dam operation rules (in both mainstream and tributaries). This paper did not mention much about the type of dams (Run-of-River, small/large reservoir, high/low dams). Each of these has different rules.

> Reply 5: and this too, a relevant issue to be clarified.The main role of nearly all of the dams in the basin is hydropower production (except few in Thailand, whose main purpose is to provide water for irrigation).  The hydropower dam operation rules were developed by Lauri et al (2012), and those aim to maximise power production. The simulated dam operation impact on discharge was confirmed to be rather accurate by Räsänen et al (2017) who compared the simulated discharge with dam operation rules in place to observed ones in the upper basin. And as reviewer points out, each of the dams has different rules, depending on the active volume of the dam, input discharge, operation of upstream dams, etc. The method used is based on an optimisation algorithm that can take all these aspects into account. See more at the papers cited here.
>
> We have briefly introduced this on the revised manuscript (lines 248-251):
>
> "Dam simulation was based on the optimisation scheme developed by Lauri et al. (2012), which calculates each dam's operating rules separately in a cascade, aiming to maximise productive outflows (i.e., outflows through the turbines), thus maximising hydro-power production."

6. Hydrological station at Neak Loeung is impacted by the tide, particularly during the dry season. Similarly, Station at Chruy Changvar is influenced by the reversed flow of the Tonle Sap Lake and tide. It is good to explain how the modelling approach of this paper overcomes these issues. Moreover, the paper should elaborate on the relationship between water level and discharge at these stations.

> Reply 6: The used flood propagation model MIKE 11 has a lower boundary at South China Sea and Gulf of Thailand is thus able to take into account the tidal impact (Table S1 in the Supplement in Triet et al 2017). The same model includes Tonle Sap Lake and thus the reversed flow was accounted for with the model. When looking at the validation results of the MIKE 11 (Table 1 in Triet et al 2017), the water level agrees very well in Prek Kdam (not far from Chruy Changvar) at Tonle Sap River to the observed water levels (NSE is 0.95, i.e. very high agreement with the observed water levels (see Table 1 in Triet et al 2017) and Neak Luong (NSE is 0.98). Therefore, we can conclude that both the tide and Tonle Sap reverse flow are well represented in MIKE 11.
>
> The water results from the IWRM-Sub model, the last in chain, line well with the observed ones too. NSE for Chroy Changvar and Neak Loeung was 0.86 and 0.85, respectively (see Table 3). The NSE for discharge in these two stations was 0.80 and 0.81, respectively, indicating very good agreement also for discharge.
>
> After careful consideration, we came to a conclusion that the current validation analyses are enough to show that our modelling scheme is able to represent the dynamics in the system (in terms of both water level and discharge) and decided not to do any further analysis to elaborate the relationship between water level and

discharge. Further, this kind of extra analysis, on top of the already rather packed article, would not add anything new towards meeting the key goals of the article.

Finally, it is just a minor comment. The spelling of the Cambodian provinces should be advised by the Cambodian co-authors. The spelling looks a bit odd, especially in Fig 7. Any changes to these names should be applied throughout the paper.

Reply 7: thanks for the note. We now harmonised the spelling of the province names as given in Fig 1.

Specific comments:

• Line 137 (Fig. 1): Boundary of the provinces cannot be clearly seen.

Reply SC1: Our colleague who produced figure 1 is no longer working on the project and has proven difficult to contact, therefore we have been unable to alter the boundaries. If the editor feels this is necessary, then we can try again to contact him to make the alterations before final publication.

• Line 172 (Fig. 2): There is no legend and scale bar for the maps. It is hard to understand the different colours of the lines, points, arrows, etc. Furthermore, the names of the key stations should be added to the maps.

Reply SC2: This figure is used as a schematic outline to show the methodology and not intended for close scrutiny or to convey information about the study area. For this, we have provided Figure 1. For clarity, we provided some more explanations on boundary condition arrows. We also cite the documents in the caption from where more information is available on tiles A and B.

• Line 243 (Table 2): for better readability, a brief description of the datasets should be provided here.

Reply SC3: all of these datasets are described in Section 2.4. If we added a description of each of the dataset to the table, it would, in our opinion, lower the readability instead of increasing it. This is, because the same dataset is used in multiple scenarios and thus, there would be a lot of repetition. We now refer to the section in the text in table caption.

• Line 276 (Table 3): Not clear the model performance was evaluated with daily or monthly values or others?

Reply SC4: The model performance was evaluated with daily values. This clarification is now added to the caption.

• Lines 291-296: Results for other stations should be included in supplementary material if they could not be presented in the paper.

> Reply SC5: The results from Kratie, Kampong Cham, and Chroy Changvar are virtually indistinguishable as they are all 50 -75 km downstream of each other without any major confluences or divergences. The difference in discharge between these points as a proportion of the total is so small as to make no impact on the modelling results. We do explain this in the text.

• Line 327 (Fig. 4): It looks like there is an inconsistency of changes in water level and discharge for Jun-Sep at Neak Loeung. This could be the impact of the tide in discharge estimate? Would you please help explain this?

> Reply SC6: we did not find any inconsistency in the results, as tiles g (absolute change) and h (relative change) agree on the direction and magnitude.

• Line 327 (Fig. 4): It is not easy to read this figure as colour bars are tiny and close to each other. It may be impossible to read when printing on monochrome. One way to tackle this issue is to reconsider a different way of visualization, i.e. separate sector development (irrigation and hydropower), climate change and cumulative change (sector and climate).

> Reply SC7: thanks for the comment; we agree. We modified the figure so that there is considerably more space for each tile and thus, we think that it is now more accessible.

• Lines 783-787 (Fig. S2 and Fig. S3): The model consistently underestimated water level at Kratie, especially low flow period – 1-1.5 m. However, the model performed well for discharge. Any explanation?

> Reply SC8: this is the upper boundary and thus, the water level might be off due to a somewhat inaccurate representation of the river morphology there. However, this does not impact on the floodplain results, as Kratie is outside the floodplain and key is that the discharge is correct, and as Table 3 shows, the NSE for discharge at Kratie indicates very good agreement with the used boundary condition and observed discharge. Also, the water levels in the floodplain agree very well with observations (see our reply 3 above).

**Reviewer 2**

I commend the authors on the review work. The suggestions of both referees have been considered with utmost care, and all points are addressed. This includes expanding the modeling framework to include the 1D flood modeling step, therefore capturing also the effects at the seawards boundary,

much expanding the scenarios contemplated, and including many more simulations. It is quite impressive that the authors have been able to carry out and document so much work in a short time. The explanation of the methods is far clearer, and Fig. 2 is excellent. Calibration and validation of model results is high quality, and figures are generally far superior now. The new version is a much more solid and superior study in many respects, and conclusions/implications can be drawn with much more confidence. What authors could not implement in the revision, e.g., a more accurate topography in parts of the analysis, is satisfactorily motivated. I recommend publishing the article, pending some minor suggestions that the authors should consider, focusing mostly on the general presentation of the study and on the methods.

> Reply: Many thanks for your positive comments on the revised version of the manuscript.

Comment 1: (L. 13) scenarios are mentioned here as though it was already clear that any scenarios were included, whereas this is not the case.

> Reply 1: We have now made it clear that we run scenarios by adding text to the introduction (line 15).
>
> *"We then ran scenarios to approximate possible conditions expected by around 2050"*

Comment 2: (L. 14-18) In those sentence there is repetition that could be substituted by some indications of in which direction climate change and hydrological development, respectively, alter discharge. That will be an obvious question still in the mind of the reader after having read the abstract.

> Reply 2: We have now taken out the unnecessary repetition and added text describing the expected impact of climate change (lines 19-21).
>
> *"Projected climate change impacts are expected to decrease dry season flows and increase wet season flows, which is opposition to the expected alterations under development scenarios that consider both hydropower and irrigation."*

Comment 3: (L. 25) the first part of the closing sentence of the abstract seems not very informative. So far the reader has not received any indication about the heterogeneity or complexity of the region, and does not have the chance to learn anything meaningful here. Similarly for the ecological fragility: it's mentioned here for the first time and not much is said about it.

> Reply 3: We have now changed the abstract text to reflect these comments, and amending the last sentence (lines 28-31).
>
> *"Our findings demonstrate the substantial changes that planned infrastructural development will have on the area, potentially impacting important ecosystems and people's livelihoods, calling for actions to mitigate these changes as well as planning potential adaptation strategies."*

Comment 4: (L 39) flooding creates damage even if short-lived. Also, please check punctuation (also on line 49).

> Reply 4: We have now removed the work 'prolonged' so that this applies to all floods. We have also changed the semi-colon to a comma in both instances.

Comment 5: (L 59) since most of those papers will be explained individually in the following, it's probably not necessary to cite them all together in that line. Also later in the paper, a bunch of studies are cited repeatedly, mostly needlessly.

> Reply 5: We have removed references to Hoang 2016, Hoang 2019, and Lauri 2012 as each of these is described in the following passages. We have also removed a number of unnecessary references, mainly to Hoang et al 2016 and 2019 from the discussion.

Comment 6: (L 60) Hoang et al 2016 present results for several stations of the Mekong. To which does this result refer?

> Reply 6: We have now stated that this result refers to stations Stung Treng and Chiang Saen.

Comment 7: (L 73) it doesn't seem obvious to the reader that "These hydrological alterations are likely to intensify when considered cumulatively". In the previous sentence you report opposing outcomes on dry season flows, so that one expects alternations to compensate each other.

> Reply 7: We agree with the reviewer that these two sentences contradict one another and so have removed the latter sentence.

Comment 8: Please check that whenever a results from previous studies is reported that evokes climate change, the scenario to which it is associated is also reported here, so the reader can evaluate if any discrepancies are attributable to different study set ups or to different scenarios.

> Reply 8: Good suggestion, thanks. We have now added that both Hoang et al (2019) and Try et al (2020) use RCP projection 8.5 (lines 74 and 81).

Comment 9: (L 186) the reader is referred to Triet et al. 2020 for the forcings of MIKE11, among which the sea level rise data used in this study. That study seems to only include a 43 cm sea level rise scenario. Is that what is used in this study, and is that appropriate for both climate scenarios included here?

> Reply 9: thanks for noting that our explanation for sea level rise was inadequate. Triet et al (2020) refers with sea level rise scenarios to a combination of climate change related sea level rise and the deltaic land subsidence. They used an average of the range estimated by Manh et al (2015), i.e. 22-63 cm. The climate change related sea level rise is taken from IPCC (2014), and is estimated for our study period to be 17-38 cm – covering all the RCP scenarios from RCP2.6 to RCP8.5. There is very little difference between the RCP scenarios (RCP4.5: 19-33 cm; RCP8.5: 22-38 cm), and thus it is justified to use the same estimate for sea level rise + deltaic land subsidence for both climate scenarios.
>
> We now state this in the revised manuscript as follows (line 240-243):
>
> "The seal level boundary condition was adjusted by 43 cm for future scenarios to account for the combined effects of sea level rise and deltaic subsidence, taken as the average of the range estimated by Manh et al (2015) i.e., 22-63 cm. This value was used for both RCP4.5 and RCP8.5 as the climate change component of sea level rise for our study period taken from IPCC (2014) is relatively consistent across RCP scenarios (RCP4.5: 19-33 cm; RCP8.5: 22-38 cm)."

Comment 10: (L 240) It is fine that the reader is referred to the previous study for further details on scenarios, but it would seem important that some more information is included also here on how the effect of the reservoir is included in the simulations. What assumptions are made about the way those 126 dams are operated? It seems plausible that based on that the peak flow lamination and the environmental flows may change massively.

> Reply 10: We have now added text that describes the assumptions used when optimising the dam rules (lines 248-251).
>
> *"Dam simulation was based on the optimisation scheme developed by Lauri et al. (2012), which calculates each dam's operating rules separately in a cascade, aiming to maximise productive outflows (i.e., outflows through the turbines), thus maximising hydro-power production."*

Comment 11: (Table 2) I find the name codes of the scenarios needlessly confusing. E.g., why sometime 'Irrigation_low' is included in the name, and other times 'LI'? why scenarios including climate change sometimes have the notation CC and sometimes not. If it's too much trouble, the authors may leave names as they are.

> Reply 11: Scenarios 2-6 consider one development activity or climate change projection in isolation, and so have expanded naming. Scenarios 7-12 combine more than one element, and so are shortened to save space. We did it this way to include as much information in the earlier (2-6) names as possible.

Comment 12: (Fig. 6) Another puzzling choice is to have the two baseline maps on a different scale than the rest of the maps here. This does not have to be changed, but I wanted to point it out in case the authors agree that this is bizarre and does not facilitate visual comparison.

> Reply 12: We thought to include slightly larger baseline maps as these convey the data that all scenarios are then judged against. But we have now amended the figure so that all the maps are the same scale.

References

Hasson, Su, Pascale, S., Lucarini, V., Böhner, J., 2016. Seasonal cycle of precipitation over major river basins in South and Southeast Asia: a review of the CMIP5 climate models data for present climate and future climate projections. Atmos. Res. 180, 42–63. https://doi.org/10.1016/j.atmosres.2016.05.008.

Hoang, L.P., Lauri, H., Kummu, M., Koponen, J., van Vliet, M.T.H., Supit, I., Leemans, R., Kabat, P. & Ludwig, F., 2016. Mekong River flow and hydrological extremes under climate change. Hydrol. Earth Syst. Sci. 20, 3027-3041. https://doi.org/10.5194/hess-20-3027-2016.

Hoang, L.P., van Vliet, M.T.H., Kummu, M., Lauri, H., Koponen, J., Supit, I., Leemans, R., Kabat, P. & Ludwig, F., 2019. The Mekong's future flows under multiple drivers: How climate change, hydropower developments and irrigation expansions drive hydrological changes. Sci. Total Environ. 649, 601-609.

Huang, Y., Wang, F., Li, Y., Cai, T., 2014. Multi-model ensemble simulation and projection in the climate change in the Mekong River Basin. Part I: temperature. Environ. Monit. Assess. 186, 7513–7523.

Lauri, H., Moel, H.d., Ward, P., Räsänen, T., Keskinen, M. & Kummu, M., 2012. Future

changes in Mekong River hydrology: impact of climate change and reservoir operation on discharge. Hydrol. Earth Syst. Sci. 16, 4603-4619. 10.5194/hess-16-4603-2012.

Piani, C., Weedon, G.P., Best, M., Gomes, S.M., Viterbo, P., Hagemann, S., Haerter, J.O., 2010. Statistical bias correction of global simulated daily precipitation and temperature for the application of hydrological models. J. Hydrol. 395, 199–215.

Portmann, F. T., Siebert, S., & Döll, P., 2010. MIRCA2000-Global monthly irrigated and rainfed crop areas around the year 2000: A new high-resolution data set for agricultural and hydrological modeling. Global Biogeochemical Cycles, 24(1). https://doi.org/10.1029/2008GB003435

Räsänen, T. A., Someth, P., Lauri, H., Koponen, J., Sarkkula, J., & Kummu, M. (2017). Observed river discharge changes due to hydropower operations in the Upper Mekong Basin. Journal of Hydrology, 545, 28–41. https://doi.org/10.1016/j.jhydrol.2016.12.023

Sillmann, J., Kharin, V.V., Zhang, X., Zwiers, F.W., Bronaugh, D., 2013. Climate extremes indices in the CMIP5 multimodel ensemble: part 1. Model evaluation in the present climate. J. Geophys. Res. Atmos. 118, 1716–1733.

Triet, N. V. K., Dung, N. V., Fujii, H., Kummu, M., Merz, B., & Apel, H., 2017. Has dyke development in the Vietnamese Mekong Delta shifted flood hazard downstream? Hydrology and Earth System Sciences, 21(8), 3991–4010. https://doi.org/10.5194/hess-21-3991-2017

Yan, D., Werners, S.E., Ludwig, F. and Huang, H.Q., 2015. Hydrological response to climate change: The Pearl River, China under diYan, D., Werners, S.E., Ludwig, F. and Huang, H.Q., 2015. Hydrological response to climate change: The Pearl River, China under different RCP scenarios. Journal of Hydrology: Regional Studies, 4, pp.228-245.